# Diffusion-based Behavior Cloning in Multi-Agent Games via Dynamic Guidance

## Abstract

In offline multi-agent imitation learning, agents are constrained to learn from static datasets without interaction, which poses challenges in generalizing across diverse behaviors. Behavior Cloning (BC), a widely used approach, models conditional actions from local observations but lacks robustness under behavioral variability. Recent diffusion-based policies have been introduced to capture diverse action distributions. However, in multi-agent environments, their iterative denoising process can accumulate errors in interactive settings, degrading performance under shifting opponent behaviors. To address these challenges, we propose Diffusion Dynamic Guidance Imitation Learning (DDGIL), a diffusion-based framework built on classifier-free guidance (CFG), which balances conditional and unconditional denoising predictions. Unlike prior methods with fixed weighting, DDGIL introduces a dynamic guidance mechanism that adaptively adjusts the weight at each denoising step, enhancing stability across different agent strategies. Empirical evaluations on competitive and cooperative benchmarks show that DDGIL achieves reliable performance. In high-fidelity sports simulations, it reproduces action strategies that closely resemble expert demonstrations while maintaining robustness against diverse opponents.

## 1 Introduction

Multi-Agent Reinforcement Learning (MARL) (Chongjie Zhang, 2010; Tabish Rashid, 2018; Deheng Ye, 2020; Jiechuan Jiang, 2023) has focused on capturing inter-agent dependencies, and effective training typically relies on reward signals obtained through repeated interactions with the environment. However, in many real-world domains such as sports analytics, dense rewards are not available, and designing explicit reward functions in sparse-reward settings is often unreliable (Jiexin Xie, 2019). Moreover, online interaction is often infeasible due to cost or data collection constraints, leaving only historical trajectories consisting of states, actions, and outcomes. These limitations motivate offline imitation learning, where policies are derived solely from demonstrations without access to reward signals or additional environment interaction.

In the offline setting, imitation learning methods, particularly behavior cloning (BC), are trained on data collected against a fixed opponent, which constrains the learned policy to specific interaction patterns. When evaluated against opponents with different behaviors, such as weaker or stylistically distinct agents, these policies often exhibit unstable performance due to ineffective actions or mistimed responses.

Recent studies have explored diffusion models for imitation learning, leveraging their ability to represent complex action distributions (Cheng Chi, 2023) in an attempt to address the limitations of BC. Existing approaches can be roughly divided into three categories. The first, often referred to as Diffusion Policy (DP) (Tim Pearce, 2023; Zhendong Wang, 2023), treats the diffusion model as the policy by conditioning primarily on the state and generating actions through multi-step denoising. As illustrated in the lower part of Figure 1, the denoising process in this class of methods typically relies solely on the observed condition, which fixes the denoising direction across steps. This rigidity makes the process prone to error accumulation, particularly in interactive environments where observations evolve with the behaviors of other agents, resulting in degraded performance. The second predicts future states through diffusion and recovers actions using an inverse dynamics model (Michael Janner, 2022; Anurag Ajay, 2023). While effective in static settings, this approach can degrade in interactive

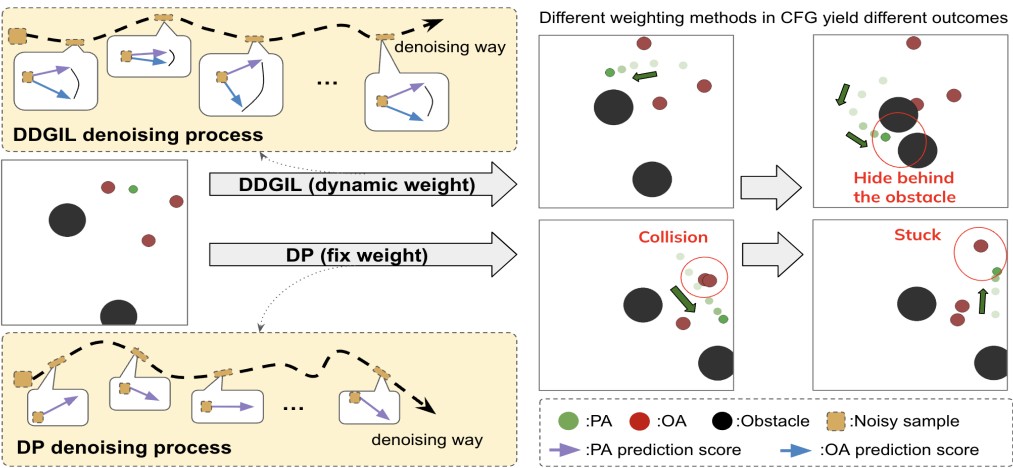

Figure 1: Tag task in the Multi-Agent Particle Environments (MPE), where three opponent agents (OA) pursue a primary agent (PA). We compare Diffusion Policy (DP, fixed weight) with DDGIL (dynamic weight). Under diverse opponent strategies, DP exhibits unstable behaviors, whereas DDGIL adapts and attains more stable performance.

domains when predicted states diverge from actual interactions. The third estimates conditional policy distributions for behavior cloning (Shang-Fu Chen, 2024), where the diffusion model provides auxiliary guidance during training. This additional signal can improve inference performance, yet the resulting policy remains essentially BC and continues to suffer from the same limitations when faced with diverse opponents. These methods indicate that diffusion-based approaches in offline imitation learning inherit key vulnerabilities of BC, highlighting the central challenge in multi-agent environments: achieving robustness under opponent variability.

Motivated by these challenges, we propose Diffusion Dynamic Guidance Imitation Learning (DDGIL), a diffusion-based imitation learning framework for offline multi-agent settings. DDGIL extends diffusion policies by introducing a minimal modification to the classifier-free guidance mechanism. As illustrated in the upper part of Figure 1, although our method employs the same denoising process as DP, the key difference lies in the use of a dynamic adjustment mechanism rather than a fixed weight. This dynamic design leverages a confidence signal to adapt to variations in the behavior of other agents, allowing the policy to remain flexible in interactive environments. Despite its simplicity, this modification improves stability and yields consistently better performance than fixed-weight diffusion policies and other baselines in both competitive and cooperative tasks.

## 2 RELATED WORK

**Multi-Agent Imitation Learning (MAIL).** Multi-Agent Imitation Learning (MAIL) has largely focused on online interaction with the environment during training (Yu et al., 2019; Nathaniel Haynam, 2025; Zare et al., 2024). Adversarial extensions such as MAGAIL (Jiaming Song, 2018) and CoDAIL (Minghuan Liu, 2020) adapt GAIL frameworks to the multi-agent setting, where a discriminator provides feedback to shape coordinated policies through interaction. While effective in capturing coordination strategies, these methods fundamentally rely on online rollouts, limiting their applicability to offline scenarios. STRIL (Shiqi Lei, 2025) represents one of the few offline approaches, filtering low-quality trajectories using strategy representations. However, it primarily focuses on modeling the agent's own behavioral heterogeneity and does not explicitly account for opponent variability. In our work, we address this gap by proposing an offline MAIL framework that adapts robustly to diverse opponent strategies.

**Diffusion Methods for Policy.** Diffusion models (Jonathan Ho, 2020) have gained significant attention for their ability to iteratively denoise Gaussian noise and generate high-quality samples. In decision-making tasks, they have been applied to imitation and reinforcement learning by treating the diffusion model as the policy itself (Anurag Ajay, 2023; Michael Janner, 2022), generating

actions or trajectories that capture multi-modal distributions. While effective in single-agent and robotic control tasks, their application to multi-agent environments remains limited. Recent work, such as MaDiff (Zhengbang Zhu, 2024), presents an offline multi-agent RL framework that uses attention-based diffusion to model coordinated behavior.

A second line of work incorporates diffusion as an auxiliary component. DiffAIL (Bingzheng Wang, 2024) augments adversarial imitation learning by adding a diffusion-based loss to the discriminator, improving its ability to distinguish expert and policy distributions. DBC (Shang-Fu Chen, 2024) is another representative offline method, which augments standard behavior cloning by combining a policy learning objective with an additional diffusion-based joint modeling loss. In both cases, diffusion serves as a supporting mechanism rather than the policy itself. In contrast, our work proposes to use diffusion models directly as policies, enabling stable and robust training in offline multi-agent imitation learning.

## 3 PRELIMINARIES

### 3.1 BEHAVIOR CLONING

Imitation learning (IL) aims to learn policies by replicating expert demonstrations without relying on explicit reward signals. In the multi-agent setting, we consider $K$ agents, where each agent $i \in \{1, \ldots, K\}$ receives an observation $s^{(i)}$ and selects an action $a^{(i)}$. We extract $(s^{(i)}, a^{(i)})$ pairs from a dataset $\mathcal{D}$. A common offline approach is behavior cloning (BC), which fits a policy by maximizing the likelihood of expert actions given the observations:

$$\pi^{(i)} = \arg\max_{\pi^{(i)}} \mathbb{E}_{(s_t^{(i)}, a_t^{(i)}) \sim \mathcal{D}} \left[ \log \pi(a_t^{(i)} \mid s_t^{(i)}) \right] \tag{1}$$

where $\pi^{(i)}$ denotes the policy of agent $i$. While effective for replicating individual behaviors, BC conditions on local observations and does not explicitly account for inter-agent coupling, which may yield suboptimal coordination in multi-agent settings.

### 3.2 DIFFUSION PROBABILISTIC MODELS

Diffusion models (DMs), particularly denoising diffusion probabilistic models (DDPM) (Jonathan Ho, 2020; Alexander Quinn Nichol, 2021), are generative models that learn to represent complex data distributions by gradually transforming Gaussian noise into structured samples through a multi-step process. We adopt this framework for action generation, allowing for high-dimensional policy modeling.

**Decision Making.** In our setting, the action $a \in \mathbb{R}^d$ is the generation target, conditioned on the current observation $s$. The forward process assumes access to state-action pairs $(s, a)$ and adds Gaussian noise to $a$ to produce a noisy sample $x_t$, defined as: $q(x_t \mid a) = \mathcal{N}\left(x_t; \sqrt{\bar{\alpha}_t} a, (1 - \bar{\alpha}_t)I\right)$, $\bar{\alpha}_t = \prod_{s=1}^{t} \alpha_s$, where $\alpha_t \in (0, 1)$ is the predefined noise schedule. To recover the original action from the noisy input, a neural network $\epsilon_\theta$ is trained to predict the added noise $\epsilon \sim \mathcal{N}(0, I)$. The noised input is computed as $x_t = \sqrt{\bar{\alpha}_t} a + \sqrt{1 - \bar{\alpha}_t} \epsilon$, with $t \sim \text{Uniform}(1, T)$. The model minimizes the following objective:

$$\mathcal{L}_{\text{DM}} = \mathbb{E}_{(s,a) \sim \mathcal{D}, \, t \sim \mathcal{U}(1,T), \, \epsilon \sim \mathcal{N}(0,I)} \left[ \|\epsilon_\theta(x_t, t \mid s) - \epsilon\|^2 \right] \tag{2}$$

At inference time, the denoising process is performed iteratively. At each reverse step $t$, the model predicts the noise $\epsilon_\theta(\mathbf{x}_t, t \mid s)$ and uses it to compute a denoised mean, which defines the mean of a Gaussian distribution used to calculate the noisy sample at the previous timestep $\mathbf{x}_{t-1}$:

$$\mu_\theta(x_t, t \mid s) = \frac{1}{\sqrt{\alpha_t}} \left( x_t - \frac{1 - \alpha_t}{\sqrt{1 - \bar{\alpha}_t}} \epsilon_\theta(x_t, t \mid s) \right), \quad x_{t-1} = \mu_\theta(x_t, t \mid s) + \sigma_t \cdot z, \quad z \sim N(0, I) \tag{3}$$

where $\sigma_t^2 = \tilde{\beta}_t = \frac{1 - \bar{\alpha}_{t-1}}{1 - \bar{\alpha}_t} (1 - \alpha_t)$, and $\sigma_t = \sqrt{\tilde{\beta}_t}$. Our imitation learning framework builds on this diffusion-based policy structure and extends it to multi-agent settings.

**Classifier-Free Guidance.** To improve conditional control during sampling, we apply classifier-free guidance (CFG) (Jonathan Ho, 2022). This method mixes the model's unconditional and conditional predictions during denoising:

$$\epsilon_{\text{CFG}} = w \cdot \epsilon_\theta(x_t, t \mid s) + (1 - w) \cdot \epsilon_\theta(x_t, t) \tag{4}$$

Here, $\epsilon_\theta(x_t \mid s)$ is the conditionally guided prediction, while $\epsilon_\theta(x_t)$ is the unconditional estimate, and $w \in [0, 1]$ is a scalar that controls the guidance strength. Throughout this paper, we adopt a convex-combination form of classifier-free guidance, mixing conditional and unconditional predictions with a weight $w \in [0, 1]$. A larger $w$ increases the adherence to the conditioning signal, while a smaller $w$ yields more diverse samples. At inference time, $\epsilon_{\text{CFG}}$ is directly substituted into the denoising update (Eq. 3) to compute the reverse steps from $x_t$ to $x_{t-1}$. Compared with classifier-based guidance, CFG avoids the difficulty of training a separate classifier under noisy inputs and has been widely observed to achieve stronger conditional fidelity and overall performance (Xi Wang & Kalogeiton, 2024; Chung et al., 2025).

## 4 METHODS

Our objective is to address the instability of behavior cloning in multi-agent interaction, particularly when agents exhibit diverse or previously unseen strategies. To this end, we propose Diffusion Dynamic Guidance Imitation Learning (DDGIL), a diffusion-based framework that combines opponent-aware score prediction with a dynamic guidance mechanism during inference. Inspired by classifier-free guidance in diffusion-based image generation (Xi Wang & Kalogeiton, 2024; Chung et al., 2025), DDGIL replaces the fixed guidance weight with a per-step adjustment derived from the confidence of conditional predictions. This mechanism reduces the limitations of fixed weighting, enabling the policy to follow conditional signals when reliable and to adapt when they are uncertain. The overall architecture is shown in Figure 2.

### 4.1 PRIMARY AGENT DIFFUSION POLICY

Our method is based on offline imitation learning with diffusion models in a multi-agent setting with $K$ agents. During training, we instantiate a separate diffusion model for each agent and designate one as the *primary agent*. The remaining agents are referred to as *opponent agents*, where the term "opponent" simply denotes agents other than the primary one: they act as adversaries in competitive tasks and as partners in cooperative tasks. The choice of the *primary agent* or *opponent agent* is user-specified during inference and does not depend on the agent's role.

To train the primary agent policy, we sample state–action pairs $(s_d^G, a_d^G)$ from the dataset $\mathcal{D}$, where $d \in 1, \ldots, |\mathcal{D}| \times H$ indexes the pair and $G$ denotes the primary agent. Here, $|\mathcal{D}|$ is the number of trajectories and $H$ their length, yielding a total of $|D| \times H$ state–action pairs.

The observation $s_d^G$ serves as the conditional input, and Gaussian noise is added to the action $a_d^G$ for diffusion model training. Following the denoising diffusion framework, we construct the input as:

$$x_t^G = \sqrt{\bar{\alpha}_t} a_d^G + \sqrt{1 - \bar{\alpha}_t} \epsilon, \quad \hat{\epsilon}_G = \epsilon_\theta^G(x_t^G, t \mid s_d^G) \tag{5}$$

where $\epsilon \sim \mathcal{N}(0, I), t \sim \text{Uniform}(1, T)$, and train a diffusion model $\epsilon_\theta^G$ to predict the noise. The training objective is defined as:

$$\mathcal{L}_{\text{DM(G)}} = \mathbb{E}_{(s_d^G, a_d^G) \sim \mathcal{D}, \, t \sim U(1,T), \, \epsilon \sim \mathcal{N}(0,I)} \left[ \|\epsilon - \hat{\epsilon}_G\|^2 \right] \tag{6}$$

This objective is equivalent to denoising score matching and promotes the original policy distribution of the primary agent as observed in the dataset. Therefore, this section focuses on modeling the behavior of the primary agent, while the modeling of the remaining opponent agents will be described in the next Section 4.2.

### 4.2 OPPONENT-AWARE DIFFUSION MODELING

We consider $K$ agents in total and write $k = K - 1$ for the number of opponents, with indices $i \in \{1, \ldots, k\}$. For each opponent $i$, we train a separate diffusion model using the same procedure

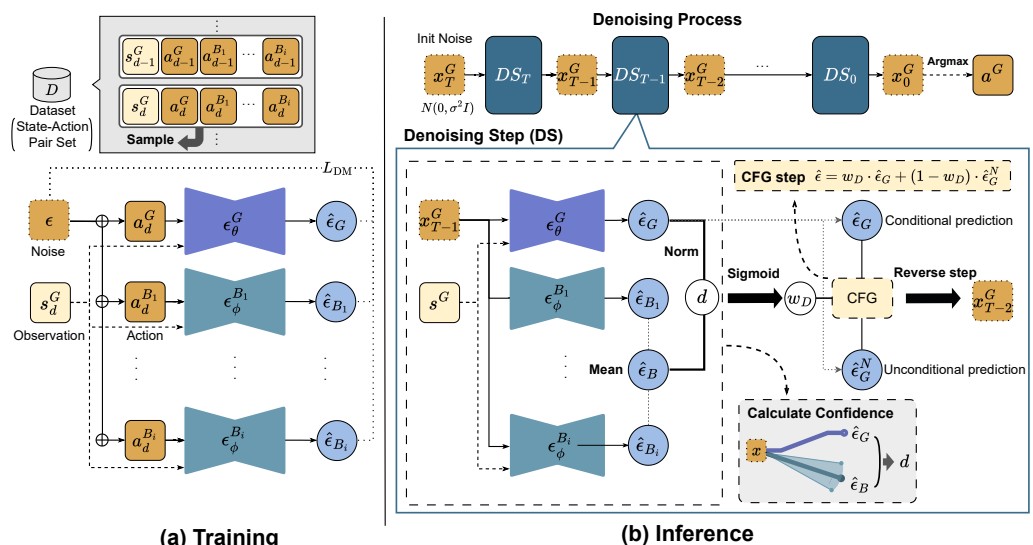

Figure 2: Overview of the **DDGIL** architecture: (a) the training pipeline, where separate diffusion-based policy models are trained for the main agent and other agents (opponents or collaborators); (b) the inference-time dynamic guidance mechanism, which computes weights at each denoising step based on the conditional scores from the primary and opponent agents, enabling adaptive policy adjustment based on the current interaction context.

as for the primary agent in Section 4.1. Concretely, we add Gaussian noise to the opponent's action $a_d^{B_i}$ from the same dataset pair $(s_d^G, a_d^G, a_d^{B_i}, \dots)$, reuse the noise sample $\epsilon$ and diffusion step $t$, and condition the model on the primary agent's observation $s_d^G$.

The noisy input and target prediction for each opponent are defined similarly to Eq. 5:

$$x_t^{B_i} = \sqrt{\bar{\alpha}_t} a_d^{B_i} + \sqrt{1 - \bar{\alpha}_t}\epsilon, \quad \hat{\epsilon}_{B_i} = \epsilon_\phi^{B_i}(x_t^{B_i}, t \mid s_d^G) \tag{7}$$

where $\epsilon_\phi^{B_i}$ predicts the diffusion noise for opponent $B_i$, conditioned on the shared context $s_d^G$. The conditioning on $s_d^G$ provides a shared reference for aligning denoising trajectories and does not imply that opponents rely on the primary agent's observation.

The overall training loss combines the reconstruction losses of the primary agent and all opponents, as shown in Figure 2(a). Since all models receive the same noise input $\epsilon$, the ideal case is that both the primary agent and its opponents predict the same noise. Sharing the same noise aligns the denoising trajectories across agents so that their disagreement reflects behavioral differences rather than randomness. To encourage this, we add a latent consistency regularization term that minimizes the difference between their predicted noise, controlled by a coefficient $c$:

$$\mathcal{L}_{\text{DM(Joint)}} = \mathbb{E}_{(s_d^G, a_d^G, \{a_d^{B_i}\}_{i=1}^{|k|}) \sim \mathcal{D}} \left[ \|\epsilon - \hat{\epsilon}_G\|^2 + \sum_{i=1}^{|k|} \|\epsilon - \hat{\epsilon}_{B_i}\|^2 + c \sum_{i=1}^{|k|} \|\hat{\epsilon}_G - \hat{\epsilon}_{B_i}\|^2 \right]. \tag{8}$$

This consistency term stabilizes the denoising dynamics under shared noise and is not intended to reduce or align behavioral differences between agents.

Recent studies show that diffusion models can reflect differences in noise through their denoising behavior (Bingzheng Wang, 2024; Mark S. Graham, 2023; Luping Liu, 2022; Yunshu Wu, 2024). Motivated by this property, we train a separate diffusion model for each opponent to capture its behavioral tendencies and decision patterns. During inference, these models do not directly generate actions; instead, their responses to shared noise inputs serve as guidance signals for the primary agent's policy. In Section 4.3, we describe how to integrate these models into an opponent-aware mechanism.

### 4.3 DYNAMIC WEIGHT ADJUSTMENT DURING INFERENCE

Each denoising step in the diffusion process updates a latent vector $x_t$, ultimately producing the final action $x_0$. We adopt the CFG framework, interpolating between conditional and unconditional predictions, and replace Eq. 4 with

$$\hat{\epsilon} = w_D \cdot \hat{\epsilon}_G + (1 - w_D) \cdot \hat{\epsilon}_G^N \tag{9}$$

where $\hat{\epsilon}_G$ and $\hat{\epsilon}_G^N$ denote conditional and unconditional predictions, respectively. Unlike the standard CFG setting that uses a fixed constant, we introduce a dynamic weight $w_D$ that adapts at each denoising step. This mechanism enables the policy to adjust its reliance on conditional information when opponent behaviors render such predictions less reliable. To compute the weight $w_D$, we measure the discrepancy between the agent's conditional prediction and the noise predicted by auxiliary opponent models. These opponent models do not perform decision making or strategic reasoning. They provide contextual score predictions that supplement the conditional estimate and are incorporated through the following measure.

$$d = \left\| \hat{\epsilon}_B - \hat{\epsilon}_G \right\|_2 \quad , \hat{\epsilon}_B = \mathbb{E}[\hat{\epsilon}_{B_i}] \quad , w_D = \sigma(d) = \frac{1}{1 + e^{-d}} \tag{10}$$

where $\hat{\epsilon}_B$ is the averaged prediction from the opponent models in the same denoising step. The value $d$ serves as a measure of confidence. Note that $d \geq 0$ guarantees $w_D \in [0.5, 1)$, so every update remains a convex combination of $\hat{\epsilon}_G$ and $\hat{\epsilon}_G^N$. This lower bound ensures the update never favors the unconditional branch, keeping conditioning as the default.

The confidence $d$ has two typical cases. When $d$ is small, the two predictors are already aligned; in this regime, increasing the weight of $\hat{\epsilon}_G$ contributes little useful signal and may amplify noise. Therefore, the rule maintains a more conservative update close to $\hat{\epsilon}_G^N$, avoiding unnecessary fluctuation. When $d$ is large, the discrepancy indicates that the step carries important predictive information, and increasing $w_D$ allows the conditional term to contribute more strongly so that this information is preserved.

Because $w_D$ is recomputed at every denoising step, the mechanism adapts naturally along the diffusion trajectory, where early steps usually provide weaker cues and later steps provide stronger ones. This dynamic adjustment improves resilience against diverse and shifting opponent strategies, avoiding the rigidity of a fixed coefficient. Section 5.3, 5.4, and 5.5 compare against fixed-weight baselines, and Appendix A further analyzes the use of $d$ as confidence and the properties of the weighting rule.

## 5 EXPERIMENTS

To evaluate the effectiveness and generality of our proposed method **DDGIL**, we evaluate on continuous, discrete, and combinatorial domains and compare against offline imitation baselines. Our experiments are designed to address the following research questions:

- **RQ1:** How well does DDGIL imitate expert behavior compared to baseline diffusion and non-diffusion methods?
- **RQ2:** Can DDGIL adapt smoothly to opponents of varying strengths, with stable behavioral and reward transitions?
- **RQ3:** How well does DDGIL perform in real-world or high-fidelity scenarios requiring strategic generalization?

### 5.1 DATA COLLECTION AND EVALUATION SETUP

**Data Collection.** For each environment, we train a task-specific reinforcement learning policy and use the best checkpoint as the expert policy $O_{\text{expert}}$ to collect an offline dataset $\mathcal{D}$ containing both successful and failed trajectories.

We additionally prepare two evaluation opponents: a mid-training policy ($O_{\text{medium}}$) and an early-stage, near-random policy ($O_{\text{weak}}$). These checkpoints are used solely for evaluation to test generalization under opponent shift and are not part of $\mathcal{D}$. The rationale for including $O_{\text{medium}}$ and $O_{\text{weak}}$ is to

introduce opponents of different strengths, which in practice also leads to noticeable variations in their behavioral patterns. All baselines are trained exclusively on $\mathcal{D}$, without using additional or lower-quality data. Further selection criteria are detailed in Appendix G.

**Environment.** We evaluate our approach on a diverse set of multi-agent environments from the PettingZoo suite (Justin K. Terry, 2020b). The benchmark covers four control tasks from the Multi-Agent Particle Environments (MPE): Tag, Push, Reference, and Spread; two pixel-based adversarial games from the Atari domain: Tennis and Boxing; and two discrete strategy games from the Classic category: Connect4 and Texas Hold'em. In addition, we introduce a custom environment, Badminton (Kuang-Da Wang, 2024b), designed to simulate realistic competitive rallies. All environments preserve their original observation and reward interfaces, with detailed descriptions provided in Appendix C.

**Evaluation metrics.** Each configuration is evaluated over 1000 episodes with five different random seeds. For *competitive tasks* (e.g., MPE-Tag, MPE-Push, Atari games, Classic games, and Badminton), we adopt the win rate as the evaluation metric, defined as $R_{\text{win}} = N_{\text{win}}/(N_{\text{win}} + N_{\text{lose}})$, where $N_{\text{win}}$ and $N_{\text{lose}}$ denote the number of wins and losses, respectively.

In tasks such as MPE-Tag and MPE-Push, a win is counted when the primary agent's episodic reward is greater than or equal to the total reward obtained by all opponents. For *cooperative tasks* (e.g., Spread, Reference), we report the average episodic reward of the primary agent, reflecting the overall team performance under the native reward attribution.

## 5.2 BASELINES

In this study, we focus on offline imitation learning and compare it against several state-of-the-art methods. Offline multi-agent imitation learning is still underexplored, especially diffusion-based approaches, so we include strong single-agent baselines for comparison, with extensions to multi-agent settings detailed in Appendix E.2. We select four representative methods:

- **Behavior Cloning (BC)**: Learns a direct state-to-action mapping using supervised learning without rewards or planning. Each agent is trained independently with a separate BC policy.
- **Diffusion Behavior Cloning (DBC)**: Combines behavior cloning with diffusion-based generation (Shang-Fu Chen, 2024), minimizing both BC loss and diffusion reconstruction loss to align actions with expert demonstrations.
- **Diffusion Policy (DP)**: Formulates the policy as a conditional diffusion model, generating actions by reversing a noise process conditioned on the current state (Tim Pearce, 2023). In our multi-agent adaptation, the model takes the primary agent's state as input and outputs actions for all agents.
- **Decision Diffusion (DD)**: Generates action sequences conditioned on state and return (Anurag Ajay, 2023). Originally designed for RL with planning, we adapt it to offline imitation learning without planners or rewards, and also extend it to the multi-agent setting with the same input–output design as DP.

We group baselines by their inference mechanism: diffusion-based methods (DP, DD) and non-diffusion ones (BC, DBC). Although DBC employs a diffusion module, its action selection follows standard behavior cloning and thus belongs to the latter group.

## 5.3 STANDARD IMITATION PERFORMANCE RESULTS

This experiment evaluates the performance of baseline models trained on the same expert demonstrations and tested against equally strong opponents $O_{\text{expert}}$. As shown in Table 1, the first row of each environment (opponent denoted as E) provides the main comparison results.

**Comparison with non-diffusion-based policy.** In environments with low-dimensional state spaces and relatively simple interaction dynamics, such as MPE, BC-based approaches maintain competitive performance. DBC, which incorporates diffusion-augmented training, achieves notable improvements on Tag. However, in domains with high-dimensional observations and stochastic transitions, such

| Env | Opp. | Non-diffusion-based | | Diffusion-based | | |
| --- | --- | --- | --- | --- | --- | --- |
| | | BC | DBC | DD | DP | DDGIL |
| Push | E | 0.79 ± 0.03 | 0.79 ± 0.02 | 0.13 ± 0.03 | 0.63 ± 0.02 | **0.81 ± 0.01** |
| | M | 0.77 ± 0.02 | 0.79 ± 0.03 | 0.21 ± 0.07 | 0.61 ± 0.03 | **0.82 ± 0.02** |
| | W | 0.78 ± 0.07 | 0.82 ± 0.06 | 0.15 ± 0.04 | 0.67 ± 0.03 | **0.84 ± 0.05** |
| Tag | E | 0.31 ± 0.09 | **0.38 ± 0.13** | 0.15 ± 0.03 | 0.15 ± 0.03 | 0.33 ± 0.08 |
| | M | 0.42 ± 0.07 | **0.47 ± 0.05** | 0.23 ± 0.06 | 0.43 ± 0.01 | 0.45 ± 0.03 |
| | W | 0.48 ± 0.02 | 0.51 ± 0.12 | 0.30 ± 0.06 | 0.54 ± 0.10 | **0.57 ± 0.06** |
| Spread | E | -11.72 ± 0.34 | -11.63 ± 0.36 | -15.03 ± 0.41 | -13.96 ± 0.25 | **-11.52 ± 0.41** |
| | M | -11.93 ± 0.51 | -11.86 ± 0.52 | -15.41 ± 0.40 | -13.83 ± 0.48 | **-11.87 ± 0.17** |
| | W | **-17.48 ± 0.44** | -17.62 ± 0.32 | -20.21 ± 0.56 | -21.01 ± 0.41 | -17.79 ± 0.47 |
| Reference | E | -30.58 ± 0.82 | -30.01 ± 1.16 | -28.68 ± 0.86 | -27.60 ± 1.02 | **-26.62 ± 1.07** |
| | M | -27.87 ± 0.49 | -30.13 ± 1.27 | -25.65 ± 0.17 | **-25.01 ± 0.45** | -26.81 ± 0.42 |
| | W | -28.46 ± 0.28 | -28.88 ± 0.27 | -32.63 ± 0.06 | -29.88 ± 0.27 | **-27.20 ± 0.18** |
| Connect4 | E | 0.14 ± 0.02 | 0.18 ± 0.03 | 0.12 ± 0.01 | 0.21 ± 0.02 | **0.26 ± 0.06** |
| | M | 0.12 ± 0.03 | 0.22 ± 0.04 | 0.13 ± 0.05 | 0.20 ± 0.03 | **0.41 ± 0.04** |
| | W | 0.41 ± 0.06 | 0.45 ± 0.03 | 0.19 ± 0.04 | 0.29 ± 0.04 | **0.47 ± 0.07** |
| Hold'em | E | 0.49 ± 0.01 | 0.53 ± 0.01 | 0.09 ± 0.02 | 0.21 ± 0.04 | **0.55 ± 0.02** |
| | M | 0.53 ± 0.03 | 0.54 ± 0.04 | 0.21 ± 0.04 | 0.35 ± 0.03 | **0.62 ± 0.02** |
| | W | 0.86 ± 0.05 | **0.88 ± 0.06** | 0.34 ± 0.04 | 0.72 ± 0.03 | **0.88 ± 0.03** |
| Tennis | E | 0.35 ± 0.07 | 0.37 ± 0.05 | 0.62 ± 0.07 | 0.72 ± 0.06 | **0.81 ± 0.05** |
| | M | 0.42 ± 0.05 | 0.46 ± 0.07 | 0.77 ± 0.05 | 0.83 ± 0.05 | **0.90 ± 0.04** |
| Boxing | E | 0.18 ± 0.03 | 0.17 ± 0.02 | 0.38 ± 0.02 | 0.43 ± 0.03 | **0.47 ± 0.03** |
| | M | 0.15 ± 0.05 | 0.17 ± 0.08 | 0.39 ± 0.06 | 0.45 ± 0.05 | **0.55 ± 0.04** |

Table 1: The win rate/average reward and standard error across different environments are computed over five seeds. Bold indicates the best result in each row. (E: Expert Opponent, M: Medium Opponent, W: Weak Opponent.)

as Atari, the performance of BC and DBC degrades considerably. For instance, on Tennis, DDGIL attains a substantially higher win rate compared to BC. The results indicate that DDGIL attains more robust performance in environments with complex observations and interaction dynamics when compared to non-diffusion-based baselines.

**Comparison with diffusion-based policy.** Compared to DP, which applies a fixed diffusion policy, DDGIL employs a dynamic guidance mechanism that adjusts conditional weighting based on confidence signals from both agent and opponent models. While DP may exhibit stable behavior in certain cases, its static weighting limits adaptability. For instance, DDGIL attains about a 10% higher win rate than DP in Tennis-Expert, highlighting the benefit of adaptive weighting. In contrast, DD relies on state prediction with inverse dynamics, which makes it prone to error accumulation in long-horizon tasks. For example, in Boxing-Expert, DDGIL achieves a relative improvement of about 24% over DD, leading to more consistent outcomes. By generating actions directly through denoising and incorporating opponent-aware feedback, DDGIL produces more stable decisions. Notably, in simpler environments such as MPE and Classic games, DDGIL also outperforms both DP and DD, underscoring that the benefit of dynamic guidance extends beyond high-dimensional settings.

## 5.4 GENERALIZATION ACROSS OPPONENT STRENGTHS

We evaluate each model against alternative opponents: $O_{medium}$ and $O_{weak}$. This setting tests whether a policy trained on strong opponents can maintain stable performance when faced with unfamiliar or weaker strategies. Results are shown in each environment's second and third rows in Table 1.

**Comparison with non-diffusion-based policy.** In MPE tasks, DDGIL performs comparably to DBC and consistently outperforms BC. In cooperative tasks, where weaker opponents act as teammates, its performance shows mixed trends: in Spread-Weak, BC and DBC achieve slightly better scores, but in Reference-Medium, DDGIL clearly surpasses both BC and DBC by 1–2 points. Overall, these results suggest that DDGIL is competitive with non-diffusion baselines and often provides stronger robustness as task difficulty increases.

| (Agent) vs Opp. | BC | DBC | DD | DP | DDGIL |
|---|---|---|---|---|---|
| (K) vs. C | $0.14 \pm 0.12$ | $0.15 \pm 0.11$ | $0.14 \pm 0.10$ | $0.43 \pm 0.06$ | $\mathbf{0.46 \pm 0.12}$ |
| (K) vs. V | $0.22 \pm 0.10$ | $0.26 \pm 0.11$ | $0.14 \pm 0.09$ | $0.61 \pm 0.11$ | $\mathbf{0.64 \pm 0.12}$ |
| (C) vs. K | $0.07 \pm 0.05$ | $0.22 \pm 0.12$ | $0.23 \pm 0.12$ | $0.39 \pm 0.04$ | $\mathbf{0.43 \pm 0.17}$ |
| (C) vs. V | $0.14 \pm 0.12$ | $0.16 \pm 0.12$ | $0.25 \pm 0.12$ | $\mathbf{0.70 \pm 0.11}$ | $0.53 \pm 0.13$ |
| (V) vs. C | $0.08 \pm 0.09$ | $0.13 \pm 0.11$ | $0.11 \pm 0.10$ | $0.35 \pm 0.07$ | $\mathbf{0.36 \pm 0.12}$ |
| (V) vs. K | $0.06 \pm 0.08$ | $0.17 \pm 0.10$ | $0.05 \pm 0.07$ | $0.36 \pm 0.08$ | $\mathbf{0.40 \pm 0.12}$ |

Table 2: Performance of baseline models in the Badminton environment. The labels K, C, and V correspond to the initials of three real-world players. In the column "(Agent) vs. Opp.", the player in parentheses denotes the controlled agent, who competes against the other two players in sequence, resulting in six matchups.

Beyond this case, DDGIL consistently outperforms baselines across Atari and Classic environments. For example, in Boxing, BC's win rate decreases when faced with weaker opponents, DBC shows only a slight improvement, while DDGIL increases by 8% points, demonstrating stronger adaptability. A similar pattern holds in Classic environments: in Connect4, BC decreases by 2% points from Expert to Medium, while DDGIL improves by 15% points.

**Comparison with diffusion-based policy.** Compared to DP and DD, DDGIL yields higher win-rate improvements against weaker opponents. While diffusion-based models are generally more robust than BC, DP and DD sometimes show non-monotonic reward changes as opponent strength decreases, like Connect4 and Push. In Push, DDGIL achieves the best results. In Boxing, DDGIL maintains consistent performance, whereas DP shows mild fluctuations. These results suggest that fixed guidance (DP) or multi-stage prediction (DD) may not adapt well to interaction shifts.

To understand these results, we analyze why DDGIL achieves stable improvements. At each step, DDGIL combines conditional and unconditional predictions with a confidence-based weight derived from their disagreement (Eqs. 9 and 10). Conditioning remains the default, ensuring the unconditional term never dominates. When disagreement is small, updates stay conservative; when large, the rule amplifies conditional guidance exactly when context matters. In contrast, a fixed weight cannot suit all steps and opponents: high values cause early overreaction, while low values weaken late responses. By recomputing the weight each step, DDGIL adapts guidance strength to observed disagreement, preserving alignment when predictions match and enhancing conditioning when they diverge. This mechanism explains the consistent gains observed across Expert, Medium, and Weak settings.

### 5.5 EVALUATION IN REAL-WORLD BADMINTON

We extend our evaluation to a virtual sports setting using the Badminton environment. All baseline models are trained on the ShuttleSet dataset (Wei-Yao Wang, 2023a;b), which contains real-world match records. We select three representative players: K, C, and V. The opponent model is Ral-lyNet (Kuang-Da Wang, 2024a), a pretrained imitation agent with diverse playing styles.

Each baseline is evaluated over 20 matches using standard scoring rules. As shown in Table 2, both DP and DDGIL rank among the top performers across all matchups. For instance, in the (K) vs. V scenario, DDGIL achieves a win rate of $0.64$, more than triple that of BC and DD. In (C) vs. V, DDGIL yields higher win rates than DBC and DD in most pairings and outperforms DP in 5 of 6 cases, with the only lower result occurring against V.

We make two observations: (1) the hybrid action space, with both discrete and continuous elements, benefits diffusion-based models due to their capacity to model multimodal outputs; and (2) DD underperforms, likely due to long-horizon prediction errors, consistent with its behavior in Atari. This result also reflects the reward-free adaptation of DD, which removes value gradients that normally guide its long-horizon optimization and can therefore weaken performance under interaction shifts. While DP remains competitive, DDGIL shows more consistent gains across most matchups. As win rate alone is insufficient, we additionally report interaction traces and trajectory-level statistics (Appendix D.3), including rally length and shot-type transitions.

## 6 CONCLUSION

We presented Diffusion Dynamic Guidance Imitation Learning (DDGIL), a diffusion-based framework for offline multi-agent imitation learning. DDGIL introduces a dynamic guidance rule that adaptively adjusts conditional and unconditional predictions during denoising, enabling stable policy generation under diverse opponent strategies without modifying training. Experiments across MPE, Atari, Classic games, and a high-fidelity badminton environment show that DDGIL outperforms diffusion and non-diffusion baselines, and in badminton it captures tactical patterns of real players, indicating potential in domains requiring strategic fidelity and adaptability. A key contribution of DDGIL is the reformulation of diffusion guidance for multi-agent interaction through opponent-aware conditional scores and an adaptive weighting mechanism, which provides a principled way to adjust guidance strength in response to varying interaction patterns.

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

# Appendix

## A   THEORETICAL INSIGHTS AND COMPARATIVE ANALYSIS

This appendix first examines the rationale of the dynamic weight computation and its mathematical consistency. We then illustrate how the mechanism balances stability across different conditions. Finally, we contrast the resulting formulation with Diffusion Policy (DP) to highlight the methodological differences between DP and DDGIL.

## A.1 VERIFICATION OF THE DYNAMIC WEIGHTING

To motivate the proposed guidance mechanism, it is necessary to clarify why $d_t = \|\hat{\epsilon}_t^B - \hat{\epsilon}_t^G\|$ with $w_D^{(t)} = \sigma(d_t)$ constitutes a reasonable choice of dynamic weight. This formulation directly links the level of disagreement between predictors to the strength of conditional guidance, ensuring that the weighting is adapted in a principled and mathematically consistent manner.

**Setup and notations.** Fix a reverse diffusion step $t$ and state $x_t$. Let $\hat{\epsilon}_t^G$ denote the conditional noise prediction (given the current condition), $\hat{\epsilon}_t^B$ the opponent prediction (e.g., averaged over opponents), and $\hat{\epsilon}_t^{NG}$ the unconditional prediction. Note that the main text used the notations $\hat{\epsilon}_G$, $\hat{\epsilon}_G^N$, and $\hat{\epsilon}_B$; here we include the timestep index $t$ explicitly for clarity of exposition. Define

$$d_t := \left\|\hat{\epsilon}_t^B - \hat{\epsilon}_t^G\right\|_2, \qquad w_D^{(t)} := \sigma(d_t) \in [0.5, 1),$$

where $\sigma(u) = \frac{1}{1+e^{-u}}$ is the logistic sigmoid. The following analysis establishes $d_t \mapsto w_D^{(t)}$ as a principled and stable dynamic weighting rule.

**(i) From noise prediction to score.** Introduce the score notation

$$s_t^G(x_t) := \nabla_{x_t} \log p_G(x_t), \qquad s_t^B(x_t) := \nabla_{x_t} \log p_B(x_t).$$

Under the DDPM parameterization, there exists a constant $c_t > 0$ (dependent only on the noise schedule) such that

$$s_t^G(x_t) \approx -c_t\, \hat{\epsilon}_t^G, \qquad s_t^B(x_t) \approx -c_t\, \hat{\epsilon}_t^B. \tag{11}$$

**(ii) $d_t$ as the magnitude of a log-likelihood ratio gradient.** Define the stepwise log-likelihood ratio

$$\ell_t(x_t) := \log \frac{p_G(x_t)}{p_B(x_t)}.$$

Differentiating with respect to $x_t$ and applying Eq. 11 gives

$$\nabla_{x_t} \ell_t(x_t) = s_t^G(x_t) - s_t^B(x_t)$$
$$\approx -c_t\big(\hat{\epsilon}_t^G - \hat{\epsilon}_t^B\big), \tag{12}$$

and thus

$$\left\|\nabla_{x_t} \ell_t(x_t)\right\| \approx c_t \left\|\hat{\epsilon}_t^B - \hat{\epsilon}_t^G\right\| = c_t\, d_t. \tag{13}$$

Hence $d_t$ (up to scale) coincides with the gradient magnitude of a discriminative objective $\ell_t$: large values indicate divergent explanations from $p_G$ and $p_B$, while small values indicate agreement.

**(iii) Smooth mapping via $\sigma(\cdot)$.** Since $d_t \geq 0$, the logistic map yields

$$w_D^{(t)} = \sigma(d_t) \in [0.5, 1),$$
$$\frac{\partial w_D^{(t)}}{\partial d_t} = \sigma(d_t)\big(1 - \sigma(d_t)\big) > 0, \tag{14}$$
$$\frac{\partial^2 w_D^{(t)}}{\partial d_t^2} = \sigma(d_t)\big(1 - \sigma(d_t)\big)\big(1 - 2\sigma(d_t)\big).$$

Thus $w_D^{(t)}$ increases monotonically with $d_t$, remains bounded within $[0.5, 1)$, and varies smoothly across steps.

**(iv) One-step consistency via line integral of $\nabla \ell_t$.** The deterministic mean displacement is

$$\Delta x_t := x_{t-1} - x_t = -\kappa_t\, \hat{\epsilon}_t, \qquad \kappa_t := \frac{1 - \alpha_t}{\sqrt{\alpha_t}\,\sqrt{1 - \bar{\alpha}_t}}, \tag{15}$$

with dynamically mixed estimate

$$\hat{\epsilon}_t = w_D^{(t)}\, \hat{\epsilon}_t^G + \big(1 - w_D^{(t)}\big)\, \hat{\epsilon}_t^{NG}. \tag{16}$$

Along the path $\gamma(\tau) = x_t + \tau \Delta x_t$, $\tau \in [0,1]$, the change in $\ell_t$ is

$$\Delta \ell_t = \int_0^1 \nabla_x \ell_t \big(\gamma(\tau)\big)^\top \gamma'(\tau) \, d\tau \; \approx \; \nabla_{x_t} \ell_t(x_t)^\top \Delta x_t \tag{17}$$

$$\stackrel{\text{(Eq.12, 15)}}{\approx} \kappa_t c_t \, (\hat{\epsilon}_t^G - \hat{\epsilon}_t^B)^\top \big[ w_D^{(t)} \hat{\epsilon}_t^G + (1 - w_D^{(t)}) \hat{\epsilon}_t^{NG} \big].$$

Expanding gives

$$\Delta \ell_t = \kappa_t c_t \Big\{ w_D^{(t)} \, (\hat{\epsilon}_t^G - \hat{\epsilon}_t^B)^\top \hat{\epsilon}_t^G + \big(1 - w_D^{(t)}\big) (\hat{\epsilon}_t^G - \hat{\epsilon}_t^B)^\top \hat{\epsilon}_t^{NG} \Big\}. \tag{18}$$

When the conditional direction aligns better with $\nabla \ell_t$ than the unconditional one, a larger $w_D^{(t)}$ yields a greater increase in $\ell_t$. Because $w_D^{(t)}$ is monotone in $d_t$ (Eq. 14) and $d_t \propto \|\nabla \ell_t\|$ (Eq. 13), the weighting adapts emphasis toward the conditional signal when discriminative evidence is strong, and away when it is weak.

**(v) Stability and compatibility.** Since $w_D^{(t)} \in [0.5, 1)$, the update in Eq. 16 remains a convex combination with conditional dominance. The derivative $\sigma'(d_t)$ in Eq. 14 bounds sensitivity to $d_t$, reducing variability due to noisy estimates. Together with Eq. 17, this shows that the rule increases $\ell_t$ proportionally to evidence while preserving the standard DDPM update form.

**Summary.** The analysis highlights four properties. First, $d_t = \|\hat{\epsilon}_t^B - \hat{\epsilon}_t^G\|$ corresponds (up to $c_t$) to the gradient magnitude of $\ell_t$ (Eq. 13), linking it to local model disagreement. Second, $w_D^{(t)} = \sigma(d_t)$ is monotone and smooth (Eq. 14), assigning higher weight under stronger evidence. Third, bounding $w_D^{(t)}$ within $[0.5, 1)$ enforces convex mixing and moderates sensitivity. This design keeps the conditional branch relatively emphasized over the unconditional component; however, while the text describes small $d_t$ as being close to the unconditional update, the lower bound of $0.5$ implies that the update only partially approaches the unconditional case. Finally, the line-integral argument (Eqs. 15–18) shows compatibility with the reverse denoising update. These properties together support $d_t$ and $w_D^{(t)}$ as a principled dynamic weighting rule.

## A.2 Relationship Between Diffusion Policy (DP) and DDGIL

This section highlights the differences between our method DDGIL and Diffusion Policy (DP). Among related works, DP is the closest to ours, as it also employs a diffusion model for behavior cloning. If the dynamic guidance mechanism is removed, the opponent model is discarded, and the guidance weight $w$ is fixed to a constant (e.g., $w_D = 1$), our method reduces to DP.

In DP, a single conditional diffusion model is trained to predict actions from observations or auxiliary information (e.g., state trajectories or history). By contrast, DDGIL introduces an auxiliary model to capture opponent behaviors and dynamically rebalances the contributions of the primary agent and opponent predictions at each denoising step, conditioned on the agent's observation. The final predicted noise is given by:

$$\textbf{DP:} \quad \hat{\epsilon} = w \cdot \hat{\epsilon}_G + (1 - w) \cdot \hat{\epsilon}_G^N, \quad w \in [0,1] \text{ (constant)} \tag{19}$$

$$\textbf{DDGIL:} \quad \hat{\epsilon} = w_D \cdot \hat{\epsilon}_G + (1 - w_D) \cdot \hat{\epsilon}_G^N, \quad w_D = \sigma(\|\hat{\epsilon}_B - \hat{\epsilon}_G\|_2) \tag{20}$$

**Key differences.** (i) DP relies on a fixed weight $w$ in Eq. 19, whereas DDGIL replaces it with a dynamic $w_D$ in Eq. 20. (ii) DDGIL increases the conditional component when the discrepancy between $\hat{\epsilon}_B$ and $\hat{\epsilon}_G$ grows ($d \uparrow \Rightarrow w_D \uparrow$), thereby prioritizing state-conditioned cues over the unconditional prior. This mitigates collapse to recurring patterns (e.g., repeated mid-court clears in Badminton) and improves robustness under opponent or style shifts. (iii) DDGIL operates purely at inference, requiring neither retraining nor architectural modification. (iv) Unlike opponent-policy models in strategic reasoning frameworks, our opponent diffusion models do not generate actions and are used only to provide contextual score predictions for guiding the primary agent.

Moreover, DP implicitly assumes that the conditional distribution $p(a \mid s)$ is sufficient for imitation, independent of interactive context. Our findings suggest that this assumption is inadequate in multi-agent environments, where strategic variability is strongly influenced by other agents. By explicitly

modeling the opponent and incorporating its predictions as an adaptive reference, DDGIL improves both robustness and behavioral fidelity. Main experiments (see Section 5.3) confirm that this dynamic formulation consistently outperforms DP, underscoring the importance of opponent-aware guidance and adaptive inference.

# B    ALGORITHM

## B.1    EMBEDDING MODEL ALGORITHM

Our overall training pipeline focuses primarily on training a diffusion model for each agent. However, in specific environments, the observed state is not a low-dimensional vector but a high-dimensional image. This differs from array-based environments such as MPE, Texas Hold'em, and Badminton, where the state can be directly fed into the model. For example, Atari environments provide $84 \times 84 \times 6$ compressed image frames, and Connect4 uses a $7 \times 6 \times 2$ binary tensor representation. Since diffusion models expect vectorized conditional inputs, using such image-based states can lead to training instability or poor convergence.

To address this, we train an auxiliary embedding model $f_{\text{emb}}$ to transform high-dimensional states into vector representations before they are used as diffusion conditions (Michelucci, 2022). This embedding model is pretrained once and kept fixed during all downstream tasks. The overall procedure is summarized in Algorithm 1. To ensure a fair comparison, we apply the same pre-trained $f_{\text{emb}}$ across all baselines, including BC, DP, DBC, and DD, in environments that involve image-based states (Atari and Connect4). The architecture and parameter settings of $f_{\text{emb}}$, as well as all other models, are detailed in Appendix E.

## B.2    DIFFUSION DYNAMIC GUIDANCE POLICY ALGORITHM

We present the dynamic guidance mechanism used during inference. For environments requiring state embeddings, a pretrained embedding model encodes state representations for training and inference. Additionally, the dynamic guidance mechanism relies on pretrained diffusion models for both the primary agent and the opponent agents. The algorithm is summarized in Algorithm 2.

---

**Algorithm 1** Training of Agent-Specific Embedding Model $f_{\text{emb}}$

---

1: **Input:** Dataset $\mathcal{D} = \{s^i\}$, encoder $E^i$, decoder $D^i$ for each agent $i$
2: **Output:** Trained encoder-decoder pairs $(E^i, D^i)$
3: **for** each agent $i \in \{1, \dots, K\}$ **do**
4:      Initialize encoder $E^i$ and decoder $D^i$
5: **end for**
6: **while** not converged **do**
7:      Sample batch of raw image states or one-hot array $\{s_t^i\}$ from $\mathcal{D}$ for all agents
8:      **for** each agent $i$ **do**
9:          **if** $s_i$ is an image **then**:
10:              Normalize input: $\tilde{s}^i \leftarrow s^i / 255.0$
11:          **else**
12:              input: $\tilde{s}^i \leftarrow s^i$
13:          **end if**
14:          Encode: $z^i \leftarrow E^i(\tilde{s}^i)$
15:          Decode: $\hat{s}^i \leftarrow D^i(z^i)$
16:          Compute reconstruction loss: $\mathcal{L}_i \leftarrow \|\hat{s}^i - \tilde{s}^i\|^2$
17:          Update $E^i$ and $D^i$ using gradient of $\mathcal{L}_i$
18:      **end for**
19: **end while**

---

---

**Algorithm 2** Inference with Dynamic Guidance

---

1: **Input:** Denoising models $\epsilon_\theta^G, \epsilon_\phi^{B_i}$, Diffusion steps $T$, Embedding model $f_{\text{emb}}$, Primary Agent index $G$, Opponent index $B$, Number of opponents $k$, Primary Agent's state $s^G$
2: **Output:** Primary agent's action $a^G$
3: **if** $s^G$ is an image **then**
4:     Embedding $z \leftarrow f_{\text{emb}}(s^G)$
5: **else**
6:     $z \leftarrow s^G$
7: **end if**
8: Initialize $x_T \sim \mathcal{N}(0, I)$
9: **for** $t = T, \ldots, 1$ **do**
10:     Predict noise with condition $\hat{\epsilon}_G \leftarrow \epsilon_\theta^G(x_t, z, t)$
11:     Predict noise without condition $\hat{\epsilon}_G^N \leftarrow \epsilon_\theta^G(x_t, \varnothing, t)$
12:     **for** each $i \in \{1, \ldots, k\}$ **do**
13:         Predict noise with condition $\hat{\epsilon}_{B_i} \leftarrow \epsilon_\phi^i(x_t, z, t)$
14:     **end for**
15:     Compute the mean of the conditional noise $\hat{\epsilon}_B = \mathbb{E}[\hat{\epsilon}_{B_i}]$
16:     Compute confidence $d \leftarrow \|\hat{\epsilon}_B - \hat{\epsilon}_G\|$
17:     Compute dynamic guidance $w_D \leftarrow \sigma(d)$                 $\triangleright \sigma(x) = \frac{1}{1+e^{-x}}$
18:     Compute epsilon $\hat{\epsilon} \leftarrow w_D \cdot \hat{\epsilon}_G + (1 - w_D) \cdot \hat{\epsilon}_G^N$
19:     $(\mu_t, \Sigma_t) \leftarrow \text{Denoise}(x_t, \hat{\epsilon})$
20:     $x_{t-1} \sim \mathcal{N}(\mu_t, \Sigma_t)$
21: **end for**
22: **Action selection:** For discrete dimensions, $a^G \leftarrow \arg\max_a (x_0[a])$; for continuous dimensions, take the real-valued output from $x_0$.

---

## C  ENVIRONMENT SETTINGS

To comprehensively evaluate the adaptability and robustness of our proposed method under diverse interaction scenarios, we select a range of representative multi-agent reinforcement learning (MARL) environments as our evaluation benchmarks. A key criterion for environment selection is a multi-agent structure; accordingly, we primarily adopt environments from the PettingZoo library (Justin K. Terry, 2020b). PettingZoo is a Python library designed explicitly for MARL research, offering a unified API and supporting various interaction types, including cooperative, competitive, and communication-based settings.

| Environment | Category | Interaction Type | Action Type |
|---|---|---|---|
| Push | MPE | Parallel | Discrete |
| Tag | MPE | Parallel | Discrete |
| Spread | MPE | Parallel | Discrete |
| Reference | MPE | Parallel | Discrete |
| Tennis | Atari | Parallel | Discrete |
| Boxing | Atari | Parallel | Discrete |
| Connect4 | Classic | AEC | Discrete |
| Texas Hold'em | Classic | AEC | Discrete |
| Badminton | Real | AEC | Discrete + Continuous |

Table 3: Metadata for each environment, including its category, interaction mode, and action space type. Badminton includes hybrid action/state features (state: 11 discrete + 6 continuous; action: 11 discrete + 4 continuous).

PettingZoo environments are categorized into two major interaction protocols: Agent Environment Cycle (AEC) and Parallel. AEC environments enforce turn-based interactions where agents observe and act sequentially, making them suitable for step-wise strategic settings. In contrast, Parallel

environments allow all agents to observe and act, simultaneously simulating real-time or synchronous interaction. These two formats differ in data collection structure, and we detail their respective recording formats in the Dataset section.

To capture diverse task structures and input modalities, we categorize the environments into four types: (1) **MPE** (Multi-agent Particle Environments), which are vector-based and emphasize coordination and adversarial interaction; (2) **Atari**, which provides pixel based inputs and complex competitive dynamics; (3) **Classic** environments, such as Connect4 and Texas Hold'em, which feature well-defined rules and game-theoretic structure; and (4) **Badminton**, a high-fidelity sports simulation inspired by real-world gameplay.

Table 3 summarizes the structural and behavioral characteristics of each domain, including category, interaction type, and action space. Additional specifications, such as state dimensionality, number of agents, action space size, and maximum episode length, are provided in Table 4, which also guides model configuration and training.

- **MPE–Tag.** This predator-prey environment involves one fast-moving good agent and three slower adversaries. The good agent incurs a penalty of -10 upon each collision with an opponent, while adversaries receive a reward of +10 for successfully hitting the good agent. The environment also includes two static obstacles blocking movement and influencing path planning. The good agents don't run to infinity, and they are also penalized for exiting the area

- **MPE–Push.** This environment consists of one good agent, one opponent, and a single landmark. The good agent receives a reward based on its proximity to the landmark, while the opponent is rewarded proportionally to the difference between its distance and the good agent's distance to the landmark. As a result, the opponent is incentivized to push the good agent away from the landmark to maximize its reward.

- **MPE–Spread.** This environment consists of $N$ agents and $N$ landmarks (with a default of $N = 3$). Agents are tasked with collectively covering all landmarks while minimizing inter-agent collisions. Globally, the team receives a shared reward based on the sum of the minimum distances from each landmark to the nearest agent. Locally, each agent incurs a penalty of $-1$ for every collision with another agent. The relative contribution of global versus local rewards is modulated by a local ratio, allowing for flexible trade-offs between cooperation and collision avoidance.

- **MPE–Reference.** This environment features two agents and three uniquely colored landmarks. Each agent aims to reach its designated target landmark, the identity of which is known only to the other agent. Both act as speakers and listeners, exchanging information to locate their targets. Local rewards are based on each agent's distance to its target, while global rewards depend on the average distance of all agents to their respective targets.

- **Atari–Tennis.** This environment is a competitive two-player game focused on positioning and prediction. Each agent aims to strike the ball past the opponent while preventing it from crossing their own side. A successful point yields a reward of $+1$ to the scorer and $-1$ to the opponent. To prevent stalling, players are penalized $-1$ if they fail to serve within 3 seconds of receiving the ball, introducing a non-zero-sum aspect to the game.

- **Atari–Boxing.** This environment simulates an adversarial boxing match emphasizing precise control and reactive strategy. Over a fixed duration of approximately 128 steps, agents can move and punch at each timestep. Scoring is based on the effectiveness of punches: 1 point for a long jab, 2 points for a close-range power punch. Each successful action yields a corresponding positive reward for the scorer and an equivalent negative reward for the opponent.

- **Classic–Connect4.** It is a two-player, turn-based game that aims to align four consecutive tokens vertically, horizontally, or diagonally on a 7-column grid. On each turn, a player drops a token into a selected column, and it falls to the lowest available position. Tokens cannot be placed in full columns. The game ends when either player achieves a sequence of four tokens or when all columns are filled, resulting in a draw if no player has won.

- **Classic–Texas Hold'em (Limit).** It is a simplified variant of Limit Texas Hold'em with two players, two betting rounds, and a deck of six cards (Jack, Queen, King in two suits). Each

player is dealt one private card, followed by a betting round, after which a single public card is revealed. A second round of betting ensues. The player with the highest-ranked hand at the end wins the game and receives a reward of $+1$, while the loser gets $-1$. At any point, a player may fold, forfeiting the game.

- **Real–World Sports: Badminton.** The Badminton environment is sourced from the CoachAI framework (Kuang-Da Wang, 2024b) and built upon the ShuttleSet dataset (Wei-Yao Wang, 2023b). ShuttleSet is currently the largest publicly available badminton singles dataset, covering 44 matches from 2018 to 2021 with over 36,000 annotated strokes. Each rally is annotated with rich metadata, including shot type, shot location, and player positions, making it especially suitable for imitation learning and tactical modeling. Each agent controls a player in a fast-paced rally, aiming to return shots and score against the opponent. This environment introduces real-world game dynamics, long-term interaction, and a more complex state-action distribution than synthetic settings.

| Environment | Agent SD | Opp. SD | Num Agent | Num Opp. | AD | Traj. Length |
|---|---|---|---|---|---|---|
| Push | 19 | 8 | 1 | 1 | 5 | 25 |
| Tag | 14 | 16 | 1 | 3 | 5 | 25 |
| Spread | 18 | – | 3 | 0 | 5 | 25 |
| Reference | 21 | – | 2 | 0 | 50 | 25 |
| Connect4 | $2\times6\times7$ | $2\times6\times7$ | 1 | 1 | 7 | 42 (21) |
| Hold'em | 72 | 72 | 1 | 1 | 4 | 50 (25) |
| Boxing | $84\times84\times6$ | $84\times84\times6$ | 1 | 1 | 18 | 128 |
| Tennis | $84\times84\times6$ | $84\times84\times6$ | 1 | 1 | 18 | 128 |
| Badminton | 17 | 17 | 1 | 1 | 15 | 60 (30) |

Table 4: Environment-specific state/action dimensions and agent settings. "Traj. Length" denotes the maximum steps per episode. "SD" and "AD" denote the abbreviations for State Dimension and Action Dimension, respectively. Values in parentheses indicate per-agent lengths in turn-based (AEC) settings, typically computed as max steps divided by the number of agents. This table supports the model architecture in Table 3.

# D  EXTRA EXPERIMENTS

This section presents an ablation study to investigate the performance differences between DDGIL and other baselines. Specifically, we examine: (i) the impact of varying the quantity of training data; (ii) the effect of using fixed weights compared to DDGIL; (iii) the adequacy of win rate in the Badminton environment, supplemented by analyses of score causes, mistake patterns, and rally length distributions to assess imitation quality; (iv) comparisons with reinforcement learning models;(v) the scalability of DDGIL to multiple agents; and (vi) the necessity of employing multiple diffusion models within DDGIL.

## D.1  EFFECT OF VARYING DATASET SIZE

We begin by analyzing the effect of data quantity on model performance. The dataset sizes in the main experiments were set to 500 episodes for MPE and Classic, and 250 for Atari. We exclude the Badminton environment from this analysis, as its dataset is fixed and does not allow for controlled variation in data volume. To evaluate sensitivity to data availability, we train each baseline with varying amounts of data: 50, 100, and 250 episodes for MPE and Classic; 10, 50, and 100 episodes for Atari. All models are trained using the same configurations as in the main experiments and evaluated against the opponent at the expert level $O_{\text{expert}}$. The results are presented in Figure 3.

The figure shows that most baseline models improve as the training data increases. BC, DBC, and DDGIL achieve competitive results in the three MPE tasks, whereas DDGIL consistently outperforms other baselines in Atari and Classic environments. Moreover, the average reward improvement across all tasks remains within 50% when comparing the smallest and largest dataset sizes. These results suggest that, despite some variations, the overall performance of our models is relatively robust with respect to the quantity of data.

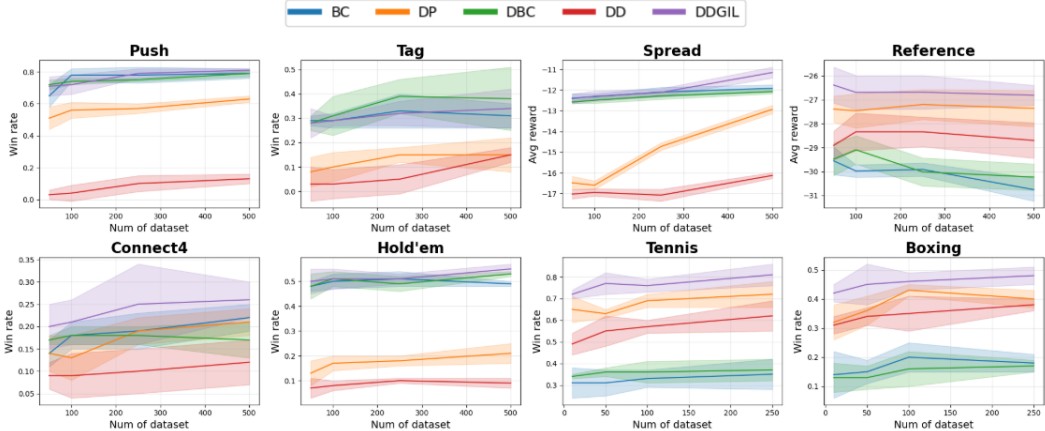

Figure 3: Dataset configurations for analyzing the impact of data quantity on model performance. MPE and Classic tasks are evaluated with 50, 100, and 250 episodes, while Atari tasks are evaluated with 10, 50, and 100 episodes.

## D.2 FIXED GUIDANCE WEIGHTS VERSUS ADAPTIVE WEIGHTS

We evaluate the effect of fixed guidance weights by setting $w$ to 1.0, 0.75, 0.5, 0.25, and compare these fixed-weight settings against the adaptive weighting used in DDGIL. Here, $w = 1.0$ corresponds to agent-only guidance (equivalent to DP), while $w = 0.0$ uses only the opponent policy. All models are evaluated with three random seeds over 300 episodes. Results are shown in Figure 4.

In MPE environments, fixing $w$ above 0.75 achieves higher rewards than DDGIL, with Push and Tag showing improvements of 0.2 to 0.3. For other tasks, $w = 1.0$ performs close to DDGIL but remains slightly lower. In Hold'em, the best performance occurs at $w = 0.5$, suggesting a balance between agent and opponent guidance benefits policy learning. Unlike Push or Tag, Hold'em relies on adapting to opponent strategies, making mixed guidance more effective than purely agent-driven decisions. When $w = 0.0$, performance consistently degrades across all tasks. Despite sharing the same noise and denoising process, the opponent policy optimizes toward opponent behavior, which conflicts with reproducing the agent's strategy. This results in a reward gap that is roughly two to three times larger than in other settings.

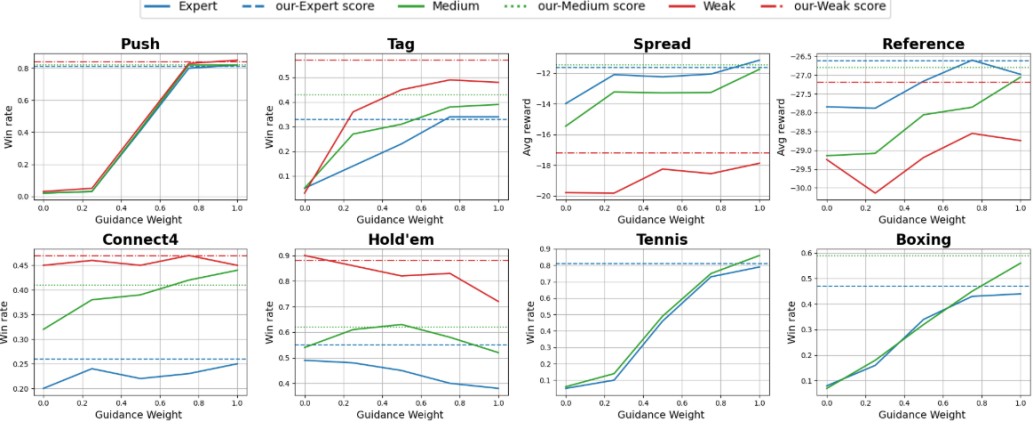

Figure 4: Performance comparison between fixed guidance weights $w$ and adaptive DDGIL. Each setting is evaluated over 300 episodes, with $w = 1.0$ representing agent-only guidance and $w = 0.0$ representing opponent-only guidance.

## D.3 ANALYSIS OF PERFORMANCE IN BADMINTON

In Section 5.5, we compared baseline models based on win rates. However, given badminton's interactive and dynamic nature, win rate alone is insufficient to evaluate how well a model replicates player behavior. To address this, we further analyze the quality of generated match processes by examining rally length distribution and score-related landing positions. The analysis uses data from 20 matches recorded during interactions with real players, following standard badminton rules, including match point settings.

**Rally Length Distribution.** Rally length reflects the tempo of exchanges and error control, serving as a key indicator of realistic gameplay. Models that fail to capture proper shot selection and defensive reactions often produce abnormally short rallies, dominated by serve or receive errors.

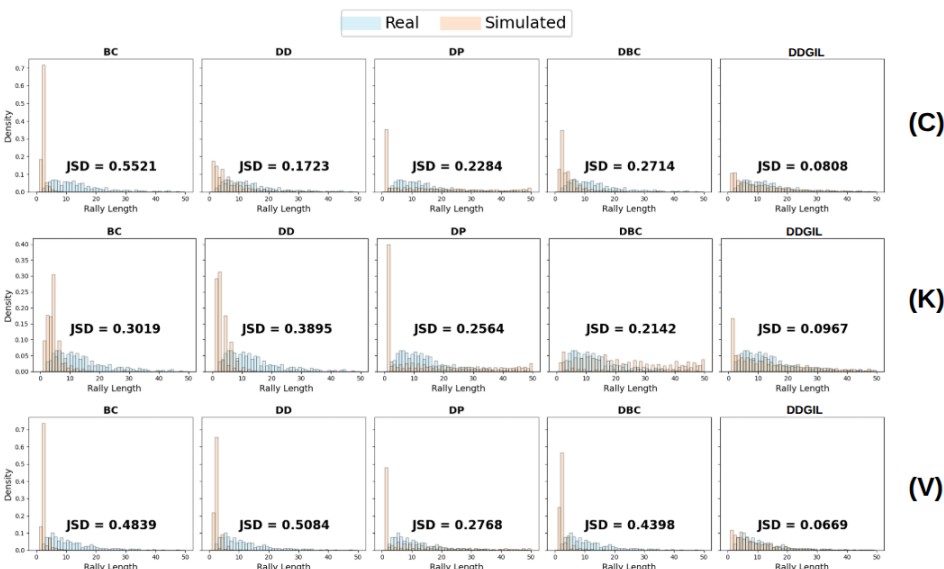

Figure 5: Comparison of rally length distributions between real players and generated simulations. Each subfigure shows the distribution for a specific player (C, K, V) across different models. JSD quantifies the difference between real and simulated distributions. DDGIL achieves the lowest JSD in all cases, indicating superior replication of realistic rally patterns.

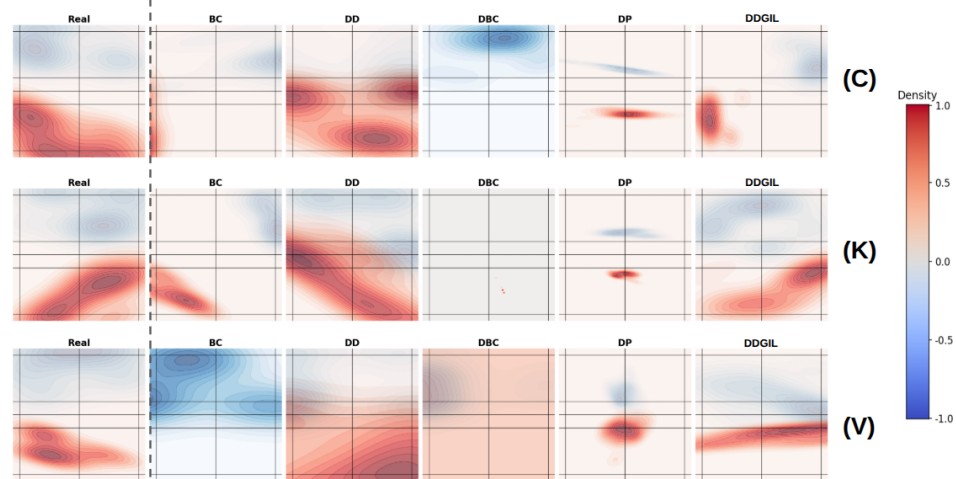

Figure 6: Landing position distributions of scoring (blue) and losing (red) shots for each player (C, K, V) across different models. Distributions are visualized using KDE heatmaps.

To evaluate this, we compute the rally length distribution for each model and measure its divergence from real data using Jensen–Shannon divergence (JSD), as shown in Figure 5. DDGIL achieves the smallest divergence, with an average JSD below 0.1, while other baselines exceed 0.2. Most alternative models show rallies concentrated between 1 to 3 shots, typically due to serve faults (length 1) or failed returns (length 2). Further inspection reveals these errors stem from incorrect shot types or boundary violations, highlighting DDGIL's superior ability to replicate realistic shot selection and play styles.

**Win/Lose landing Distribution.** We also analyze the landing positions of scoring and losing shots. Specifically, we record the positions where a point is won (on the opponent's court) and where a point is lost (on the agent's court), excluding out-of-bounds shots. This assesses whether the model captures the player's preferred attack zones and common defensive weaknesses. Alignment in scoring positions indicates learned offensive tendencies, while consistency in losing positions reflects an understanding of a player's typical weaknesses.

Figure 6 presents the comparison, with red areas denoting losing shot distributions and the blue regions for scoring shots. Though no numerical metrics are provided, KDE visualizations show that DDGIL's distributions closely match those of the real players, particularly for players K and C. For player V, the alignment is less precise but still reasonable. In contrast, BC and DBC exhibit imbalanced patterns, often lacking losing shot distributions due to frequent serve faults and out-of-bounds hits, which are not reflected in the landing statistics. DP and DD show distinct behaviors: DD produces overly dispersed landing positions, lacking clear attack patterns, while DP focuses on fixed regions for both scoring and losing shots, indicating limited adaptability. These observations further demonstrate DDGIL's advantage in mimicking player-specific strategies and behaviors.

## D.4 COMPARE WITH REINFORCEMENT LEARNING MODELS

In this work, although our primary setting is reward-free, we additionally compare DDGIL with offline RL methods to assess its generalization and stability. For a fair comparison, we recollected datasets with reward information and included IDQL (Philippe Hansen-Estruch, 2023) and OMAR (Ling Pan, 2022) as baselines. All evaluations follow the same protocol as in the main experiments under Expert, Medium, and Weak opponent conditions.

Table 5 reports the comparison between DDGIL and offline RL baselines. On MPE tasks, DDGIL achieves consistent gains, with win rates on Push and Tag exceeding IDQL and OMAR by up to +6–8% under certain opponent conditions. In Spread, DDGIL remains competitive but trails the strongest baseline by roughly 2–3%. In Classic and Atari domains, RL methods generally dominate: in Connect4 and Tennis, DDGIL is lower by 5–10%, while in Boxing the gap narrows to within 2–3%. For Hold'em, DDGIL surpasses RL baselines under Weak.

Overall, these results indicate that DDGIL is particularly effective in MPE environments, where coordination and adaptation to opponent strategies are critical. While RL baselines retain an advantage in Connect4 and most Atari domains, DDGIL demonstrates competitive performance in reward-free settings and achieves superior outcomes in interaction-heavy tasks such as Push, Tag, and Expert Hold'em.

## D.5 SCALABILITY OF MULTIPLE AGENTS FOR DDGIL

To evaluate the scalability of DDGIL with respect to the number of opponents, we conducted experiments on the MPE Tag task with 3, 6, and 10 opponents. Each configuration was evaluated in a fixed expert setting for 100 episodes, averaging the results in three random seeds with standard deviations, as reported in Table 6. In addition, single-step inference time and memory usage were recorded.

Results show that DDGIL maintains stable win rates as the number of opponents increases. With three opponents, DDGIL achieves a win rate of 0.34, which is lower than DBC. As the number of opponents increases to six and ten, DDGIL maintains the highest performance, whereas BC and DBC degrade more substantially. In contrast, DD and DP yield considerably lower win rates across all configurations, underscoring their limited adaptability in multi-opponent settings.

| Env | Opp. | RL methods | | IL methods |
| --- | --- | --- | --- | --- |
| | | IDQL | OMAR | DDGIL |
| Push | E | **0.81 ± 0.04** | 0.80 ± 0.04 | **0.81 ± 0.01** |
| | M | **0.86 ± 0.03** | 0.84 ± 0.05 | 0.82 ± 0.02 |
| | W | 0.83 ± 0.06 | 0.77 ± 0.04 | **0.84 ± 0.05** |
| Tag | E | 0.33 ± 0.08 | 0.33 ± 0.05 | **0.34 ± 0.08** |
| | M | **0.45 ± 0.03** | 0.39 ± 0.06 | **0.45 ± 0.03** |
| | W | **0.60 ± 0.07** | 0.58 ± 0.04 | 0.57 ± 0.06 |
| Spread | E | **-10.68 ± 0.25** | -12.54 ± 0.06 | -11.62 ± 0.41 |
| | M | -11.91 ± 0.42 | -13.93 ± 0.08 | **-11.87 ± 0.17** |
| | W | **-15.67 ± 0.33** | -15.82 ± 0.10 | -17.79 ± 0.47 |
| Connect4 | E | 0.31 ± 0.02 | **0.32 ± 0.04** | 0.26 ± 0.04 |
| | M | **0.50 ± 0.03** | 0.47 ± 0.04 | 0.41 ± 0.06 |
| | W | **0.54 ± 0.02** | 0.53 ± 0.03 | 0.47 ± 0.06 |
| Hold'em | E | **0.57 ± 0.04** | 0.53 ± 0.02 | 0.55 ± 0.02 |
| | M | 0.61 ± 0.03 | **0.67 ± 0.01** | 0.62 ± 0.02 |
| | W | 0.83 ± 0.07 | 0.85 ± 0.03 | **0.88 ± 0.04** |
| Tennis | E | **0.85 ± 0.02** | 0.84 ± 0.04 | 0.81 ± 0.05 |
| | M | 0.89 ± 0.05 | **0.94 ± 0.06** | 0.90 ± 0.04 |
| Boxing | E | **0.53 ± 0.05** | 0.47 ± 0.06 | 0.48 ± 0.03 |
| | M | **0.66 ± 0.04** | 0.65 ± 0.07 | 0.52 ± 0.04 |

Table 5: Comparison between RL methods (IDQL, OMAR) and IL method (DDGIL). Results show that DDGI outperforms IDQL and OMAR in some environments.

| Opp. Num | Baseline | Win rate | Time / step | Memory |
| --- | --- | --- | --- | --- |
| **3** | DDGIL | 0.34 ± 0.08 | 61.3 ms | 1.16 GB |
| | BC | 0.31 ± 0.05 | 4.0 ms | 0.31 GB |
| | DBC | **0.38 ± 0.13** | 4.1 ms | 0.33 GB |
| | DD | 0.15 ± 0.03 | 46.4 ms | 0.45 GB |
| | DP | 0.15 ± 0.03 | 41.1 ms | 0.45 GB |
| **6** | DDGIL | **0.37 ± 0.06** | 72.7 ms | 1.16 GB |
| | BC | 0.32 ± 0.09 | 4.5 ms | 0.31 GB |
| | DBC | 0.32 ± 0.07 | 4.5 ms | 0.33 GB |
| | DD | 0.13 ± 0.05 | 51.2 ms | 0.45 GB |
| | DP | 0.15 ± 0.07 | 48.9 ms | 0.45 GB |
| **10** | DDGIL | **0.35 ± 0.05** | 95.2 ms | 1.54 GB |
| | BC | 0.28 ± 0.04 | 6.4 ms | 0.33 GB |
| | DBC | 0.31 ± 0.05 | 6.3 ms | 0.35 GB |
| | DD | 0.13 ± 0.08 | 57.1 ms | 0.47 GB |
| | DP | 0.12 ± 0.08 | 56.8 ms | 0.46 GB |

Table 6: Win rate and computational cost with increasing opponent counts in MPE-Tag.

In terms of computational cost, DDGIL incurs a higher inference time and memory usage due to the additional opponent modeling. For example, the inference time increases from 121.3 ms with 3 opponents to 251.2 ms with 10 opponents, and the memory usage increases from 1.16 GB to 1.54 GB. Although these values are larger than BC and DBC, they remain within the range of other diffusion-based baselines (DD, DP).

Overall, DDGIL demonstrates better performance scalability than BC, DBC, DD, and DP as the number of opponents increases, while its computational cost increases predictably with task complexity. This indicates that the dynamic guidance mechanism effectively stabilizes learning under more challenging multi-opponent environments.

## D.6 NECESSITY OF MULTIPLE DIFFUSION MODELS IN DDGIL

In the original design, each agent is assigned an independent diffusion model, with separate conditional generation and dynamic guidance. To examine the necessity of this multi-model setup, we conducted an ablation by replacing it with a single shared model. The shared model takes the primary agent's observation as input and outputs actions for all agents through a shared encoder and multi-head denoisers, while retaining dynamic guidance during inference. Evaluation metrics include average return, stability, and computational cost.

| Env | Reward/WR | Origin Reward/WR | Time | Origin Time | Memory | Origin Mem. |
|---|---|---|---|---|---|---|
| Push | $0.13 \pm 0.03$ | $0.81 \pm 0.02$ | 39.3 ms | 47.1 ms | $0.48 \pm 0.00$ | $0.59 \pm 0.08$ |
| Tag | $0.05 \pm 0.10$ | $0.33 \pm 0.08$ | 48.4 ms | 61.3 ms | $0.56 \pm 0.13$ | $0.94 \pm 0.11$ |
| Spread | $-22.68 \pm 0.88$ | $-11.52 \pm 0.41$ | 41.6 ms | 62.4 ms | $0.52 \pm 0.07$ | $0.65 \pm 0.07$ |
| Reference | $-38.82 \pm 1.07$ | $-26.62 \pm 1.07$ | 35.2 ms | 46.2 ms | $0.50 \pm 0.08$ | $0.58 \pm 0.12$ |
| Tennis | $0.46 \pm 0.12$ | $0.81 \pm 0.05$ | 36.5 ms | 42.2 ms | $0.59 \pm 0.07$ | $0.59 \pm 0.01$ |
| Boxing | $0.11 \pm 0.09$ | $0.47 \pm 0.03$ | 35.8 ms | 44.5 ms | $0.51 \pm 0.09$ | $0.60 \pm 0.05$ |
| Connect4 | $0.15 \pm 0.11$ | $0.26 \pm 0.04$ | 37.1 ms | 45.2 ms | $0.47 \pm 0.05$ | $0.57 \pm 0.08$ |
| Hold'em | $0.34 \pm 0.42$ | $0.55 \pm 0.02$ | 33.0 ms | 45.3 ms | $0.47 \pm 0.02$ | $0.57 \pm 0.04$ |

Table 7: When DDGIL is implemented as a single shared model, compared with the original multi-model setup. *Origin* denotes the performance of the original model configuration. **WR** is the abbreviation for *Win Rate*.

Table 7 shows that the shared model reduces computational cost, achieving on average 26% shorter inference time (e.g., 48.4 ms vs. 61.3 ms in Tag) and about 17% lower memory usage. However, this efficiency gain comes with performance degradation in most environments. In Push, Tag, Tennis, and Boxing, win rates decline notably, while in Spread and Reference, rewards drop by more than 10 points. In contrast, Connect4 and Hold'em show smaller differences, indicating less sensitivity to model sharing.

The performance gap arises because, in the multi-model setting, the opponent model serves only as an external reference for condition and does not affect the gradient updates of the primary agent. In the shared architecture, all agents are generated by the same model, causing gradient interference, reducing the distinctiveness of guidance vectors, and hindering convergence. In addition, changes in the denoising path primary agent indirectly alter the output of other agents, breaking the design principle of using separate models to capture interaction-specific semantics through dynamic guidance.

Therefore, although the multi-model design increases computational cost, it plays a critical role in maintaining stability and controllability in imitation learning. It also preserves separable representations of the primary agent and opponent policies, thereby enhancing the interpretability and responsiveness of dynamic guidance.

# E  MODEL ARCHITECTURE

## E.1  EXPERT REINFORCEMENT LEARNING MODEL

We configure an expert strategy model for each environment to ensure the quality and consistency of demonstration data used during the imitation learning phase. These expert models are designed to fully capture the rules and decision-making structure of the corresponding environment and remain fixed during data collection, without being updated jointly with the student policy. The architecture and training protocol of each expert differ according to the nature of the environment, as described below:

- **Atari (Tennis, Boxing)**: We adopt the *Multi-Agent PPO (MAPPO)* (Logan Engstrom, 2020; John Schulman, 2017) implementation from *CleanRL* (Shengyi Huang, 2022), integrated with preprocessing utilities provided by Supersuit (Justin K. Terry, 2020a). Observations from PettingZoo are cropped, normalized, and stacked into multi-channel frames. The policy network simultaneously processes both agents' observations and outputs their respective action distributions, enabling stable learning of pixel-level competitive behavior.

- **MPE (Tag, Push, Spread, Reference)**: We employ the *MADDPG* (Ryan Lowe, 2017) algorithm provided by *AgileRL* (Ustaran-Anderegg et al.), which uses a centralized Q-critic to evaluate joint action values and decentralized actors for each agent. This design effectively captures both cooperative and adversarial patterns in the multi-agent particle environment.

- **Connect4**: The expert is instantiated using the *AgileRL DQN* (Volodymyr Mnih, 2013) model with publicly available pretrained weights. These weights are obtained through

curriculum learning and self-play, enabling the generation of competent gameplay demonstrations without additional training cost.

- **Texas Hold'em**: For this imperfect-information game, we use the *Neural Fictitious Self-Play (NFSP)* (Johannes Heinrich, 2016) implementation from *RLCard* (Zha et al., 2020). NFSP maintains both a best-response policy and an average strategy memory, progressively converging to a Nash equilibrium through self-play. This allows the expert to model strategic inference over hidden information.

- **Badminton**: The expert model is based on *RallyNet* (Kuang-Da Wang, 2024a), a pretrained imitation learning model derived from real-world badminton match footage. It is capable of predicting high-quality shot sequences and footwork trajectories, providing fluent and realistic expert demonstrations.

During the expert training phase, we log the model weights across training epochs along with the corresponding evaluation rewards. These metrics serve as the basis for ranking and selecting expert strengths (e.g., medium, weak) for future ablation studies. The weight checkpoint that achieves the highest evaluation reward, typically the one saved in the final epoch, is selected as the expert policy. All subsequent rollout datasets used for offline imitation learning are generated via interaction between this selected expert and the environment.

### E.2 BASELINE MODEL

To benchmark the proposed framework under the offline imitation learning setting, we construct a set of baseline models. Our survey of existing literature indicates that most multi-agent imitation learning methods are designed for online training and are thus incompatible with our offline setting. We therefore adopt single-agent offline imitation learning algorithms as the primary baselines, and additionally include two multi-agent offline reinforcement learning algorithms as supplementary experiments in Appendix D.5.

- **BC**: A classical behavior cloning model that employs a three-layer MLP with ReLU activations. It directly learns to map states to actions in a single-stage training procedure.

- **DBC**: Based on the Diffusion Model-Augmented Behavioral Cloning (Shang-Fu Chen, 2024), this model includes a two-stage training process. A diffusion model is trained to learn improved representations, followed by a behavior cloning policy. The decision component is identical to the BC architecture.

- **DP**: Implemented using the *Clean Diffuser* (Zibin Dong, 2024), Diffusion Policy (Tim Pearce, 2023) adopts a denoising diffusion probabilistic model (DDPM). A DIT1D-based UNet is used as the diffusion backbone, while MLPCondition is applied for conditional state input. The model predicts a one-step action conditioned on a short state sequence. Originally designed for single-agent decision making, we extend it to the multi-agent setting by conditioning on the primary agent's state and generating action distributions for all agents, from which the primary agent's action is taken, like a shared-model design with multi-head outputs.

- **DD**: We adapt Decision Diffuser (Anurag Ajay, 2023) into an offline imitation learning formulation using the *Clean Diffuser*. In contrast to its original reinforcement learning design, our implementation removes reward-based conditioning and relies solely on state information. Since each state transition in a multi-agent environment is influenced by all agents' actions, modeling individual-agent trajectories in isolation is insufficient. To address this, we introduce an inverse dynamics module that predicts the actions of all agents given consecutive states. The inverse model outputs a vector of action dimension action × number of agents, enabling accurate recovery of interaction patterns across agents.

- **IDQL**: IDQL (Philippe Hansen-Estruch, 2023) adopts a generalized IQL architecture consisting of a Q-function (critic) trained solely on dataset actions and a diffusion-parameterized behavior policy (actor). The actor generates samples from the diffusion model and applies importance sampling with weights computed from the critic to obtain the final policy. Our implementation is based on the *Clean Diffuser* to support training and inference of diffusion-based behavior policies.

| Method | Param | Push | Tag | Spread | Reference | Connect4 | Hold'em | Tennis | Boxing | Badminton |
|---|---|---|---|---|---|---|---|---|---|---|
| **BC** | in dim | [8, 19] | [14, 16] | 18 | 21 | 64 | 72 | 64 | 64 | 17 |
| | out dim | 5 | 5 | 5 | 50 | 7 | 4 | 18 | 18 | 15 |
| | hid dim | 256 | 256 | 256 | 256 | 256 | 256 | 256 | 256 | 256 |
| **DBC** | **Diffusion** | | | | | | | | | |
| | in dim | [13, 24] | [19, 21] | 23 | [71, 71] | 71 | 76 | 82 | 82 | 32 |
| | out dim | [13, 24] | [19, 21] | 23 | [71, 71] | 71 | 76 | 82 | 82 | 32 |
| | hid dim | 256 | 256 | 256 | 256 | 256 | 256 | 256 | 256 | 256 |
| | step | 1000 | 1000 | 1000 | 1000 | 1000 | 1000 | 1000 | 1000 | 1000 |
| | **BC** | | | | | | | | | |
| | in dim | [8, 19] | [14, 16] | 18 | 21 | 64 | 72 | 64 | 64 | 17 |
| | out dim | 5 | 5 | 5 | 50 | 7 | 4 | 18 | 18 | 15 |
| | hid dim | 256 | 256 | 256 | 256 | 256 | 256 | 256 | 256 | 256 |
| **DP** | in dim | [8, 19]×2 | [14, 16]×2 | 18×2 | 21×2 | 64×2 | 72×2 | 64×2 | 64×2 | 17×2 |
| | out dim | 5 | 5 | 5 | 50 | 7 | 4 | 18 | 18 | 15 |
| | hid dim | 384 | 384 | 384 | 384 | 384 | 384 | 384 | 384 | 384 |
| | step | 15 | 15 | 15 | 15 | 15 | 15 | 20 | 20 | 20 |
| | ex step | 5 | 5 | 5 | 5 | 5 | 5 | 8 | 8 | 5 |
| **DD** | **Diffusion** | | | | | | | | | |
| | in dim | [8, 19]×H | [14, 16]×H | 18×H | 21×H | 64×H | 72×H | 64×H | 64×H | 17×H |
| | out dim | [8, 19]×H | [14, 16]×H | 18×H | 21×H | 64×H | 72×H | 64×H | 64×H | 17×H |
| | hid dim | 320 | 320 | 320 | 320 | 320 | 320 | 320 | 320 | 320 |
| | step | 20 | 20 | 20 | 20 | 20 | 20 | 25 | 25 | 25 |
| | **InvDyn** | | | | | | | | | |
| | in dim | [8, 19]×2 | [14, 16]×2 | 18×2 | 21×2 | 64×2 | 72×2 | 64×2 | 64×2 | 17×2 |
| | out dim | 5×n | 5×n | 5×n | 50×n | 7×n | 4×n | 18×n | 18×n | 15×n |
| | hid dim | 512 | 512 | 512 | 512 | 512 | 512 | 512 | 512 | 512 |
| **IDQL** | in dim | [8, 19] | [14, 16] | 18 | 21 | 64 | 72 | 64 | 64 | 17 |
| | out dim | 5×n | 5×n | 5×n | 50×n | 7×n | 4×n | 18×n | 18×n | 15×n |
| | hid dim | 512 | 512 | 512 | 512 | 512 | 512 | 512 | 512 | 512 |
| **OMAR** | in dim | [8, 19] | [14, 16] | 18 | 21 | 64 | 72 | 64 | 64 | 17 |
| | out dim | 5 | 5 | 5 | 50 | 7 | 4 | 18 | 18 | 15 |
| | hid dim | 256 | 256 | 256 | 256 | 256 | 256 | 256 | 256 | 256 |
| **DDGIL** | **EM** | | | | | | | | | |
| | in dim | - | - | - | - | 7×6×2 | - | 84×84×6 | 84×84×6 | - |
| | emb dim | - | - | - | - | 64 | - | 64 | 64 | - |
| | hid dim | - | - | - | - | 256 | - | 256 | 256 | - |
| | **Diffusion** | | | | | | | | | |
| | in dim | [8, 19] | [14, 16] | 18 | 21 | 72 | 64 | 64 | 64 | 17 |
| | out dim | 5 | 5 | 5 | 50 | 7 | 4 | 18 | 18 | 15 |
| | hid dim | 256 | 256 | 256 | 256 | 256 | 256 | 256 | 256 | 256 |
| | step | 15 | 15 | 15 | 15 | 15 | 15 | 20 | 20 | 20 |

Table 8: Model Architecture Parameters for baseline. Hyperparameter configurations for all baseline models and our proposed DDGI across seven environments.

- **OMAR**: Offline Multi-Agent RL with Actor Rectification (OMAR) (Ling Pan, 2022) combines first-order policy gradients with zeroth-order optimization to address the non-concavity of conservative value functions in the actor parameter space, reducing the risk of suboptimal convergence. This design mitigates global coordination failures caused by suboptimal policies of individual agents in offline multi-agent reinforcement learning. Our implementation follows the original OMAR architecture to support policy optimization in multi-agent control tasks.

The hyperparameter settings for all baseline models are summarized in Table 8. For clarity, we define the following abbreviations: **in dim** (input dimension), **hid dim** (hidden dimension), **out dim** (output dimension), **step** (diffusion sampling steps), **ex step** (extra steps in DP), **EM** (embedding model in DDGIL), and **InvDyn** (inverse dynamics module in DD). In DD, **H** is the trajectory horizon (defaulting to the environment's episode length), while **n** denotes the number of agents. For Push/Tag, state dimensions are shown as (agent, opponent). In Tennis, Boxing, and Connect4, state dimensions follow DDGIL's embedding size.

| Method | Param | Push | Tag | Spread | Reference | Connect4 | Hold'em | Tennis | Boxing | Badminton |
|---|---|---|---|---|---|---|---|---|---|---|
| - | bs | 256 | 256 | 256 | 256 | 256 | 256 | 64 | 64 | 256 |
| **BC** | lr | 1e-4 | 1e-4 | 1e-4 | 1e-4 | 1e-4 | 1e-4 | 1e-4 | 1e-4 | 1e-4 |
| | epoch | 300 | 300 | 300 | 300 | 300 | 300 | 400 | 400 | 300 |
| **DBC** | **Diffusion** | | | | | | | | | |
| | lr | 5e-4 | 5e-4 | 5e-4 | 5e-4 | 5e-4 | 5e-4 | 1e-5 | 1e-5 | 5e-4 |
| | epoch | 1000 | 1000 | 1000 | 1000 | 1000 | 1000 | 1000 | 1000 | 1000 |
| | **BC** | | | | | | | | | |
| | lr | 5e-4 | 5e-4 | 5e-4 | 5e-4 | 5e-4 | 5e-4 | 1e-5 | 1e-5 | 5e-4 |
| | epoch | 300 | 300 | 300 | 300 | 300 | 300 | 400 | 400 | 300 |
| **DP** | lr | 5e-4 | 5e-4 | 5e-4 | 5e-4 | 5e-4 | 5e-4 | 1e-4 | 1e-4 | 5e-4 |
| | epoch | 1000 | 1000 | 1000 | 1000 | 1000 | 1000 | 1200 | 1200 | 1000 |
| **DD** | **Diffusion** | | | | | | | | | |
| | lr | 5e-4 | 5e-4 | 5e-4 | 5e-4 | 5e-4 | 5e-4 | 1e-5 | 1e-5 | 5e-4 |
| | epoch | 5000 | 5000 | 5000 | 5000 | 5000 | 5000 | 8000 | 8000 | 6000 |
| | **InvDyn** | | | | | | | | | |
| | lr | 5e-4 | 5e-4 | 5e-4 | 5e-4 | 5e-4 | 5e-4 | 1e-5 | 1e-5 | 5e-4 |
| | epoch | 2000 | 2000 | 2000 | 2000 | 2000 | 2000 | 3000 | 3000 | 2000 |
| **DQL** | lr | 5e-4 | 5e-4 | 5e-4 | 5e-4 | 5e-4 | 5e-4 | 5e-4 | 5e-4 | 5e-4 |
| | epoch | 1000 | 1000 | 1000 | 1000 | 1000 | 1000 | 1000 | 1000 | 1000 |
| **OMAR** | lr | 1e-5 | 1e-5 | 1e-5 | 1e-5 | 1e-5 | 1e-5 | 1e-5 | 1e-5 | 1e-5 |
| | epoch | 5000 | 5000 | 5000 | 5000 | 5000 | 5000 | 5000 | 5000 | 5000 |
| **DDGIL** | **EM** | | | | | | | | | |
| | lr | - | - | - | - | 5e-4 | - | 1e-5 | 1e-5 | - |
| | epoch | - | - | - | - | 100 | - | 100 | 100 | - |
| | **Diffusion** | | | | | | | | | |
| | lr | 1e-5 | 1e-5 | 1e-5 | 1e-5 | 1e-5 | 1e-5 | 1e-5 | 1e-5 | 1e-5 |
| | epoch | 500 | 500 | 500 | 500 | 500 | 500 | 600 | 600 | 500 |

Table 9: Model Training Parameters for baseline in each environment.

Table 9 summarizes the hyperparameter configurations for all baseline models. The following abbreviations are used: **lr** (learning rate), **bs** (batch size), and **epoch** (total training epochs). All baselines are optimized using Adam. The batch size for each environment is listed at the top of the table.

E.3    DDGIL MODEL

To handle the diversity of state representations across environments, we structure DDGIL into two modular yet interdependent components: an embedding module and a diffusion-based policy generation module. The former transforms high-dimensional, non-vector observations into compact latent embeddings, while the latter serves as the core mechanism for skill-conditioned policy generation.

**Embedding Module.**    The embedding module is responsible for preprocessing non-vectorial state inputs, such as images or structured tensors, and converting them into fixed-length latent representations suitable for conditioning the diffusion model. The training algorithm can be found in Appendix B. We implement two variants of the embedding module tailored to different input types:

- For **Atari** (e.g., Boxing, Tennis), the encoder consists of three Conv2D layers, and the decoder mirrors this with three ConvTranspose2D layers. The raw input of shape $84 \times 84 \times 6$ is normalized before being encoded into a 64-dimensional latent vector.

- For **Connect4** The state is a $7 \times 6 \times 2$ one-hot tensor, we adopt a lightweight MLP encoder comprising two linear layers with nonlinear activations, along with a symmetric two-layer MLP decoder. No normalization is applied to the input, as it is already structured and non-visual.

For vector-based environments such as MPE and Texas Hold'em, no embedding is required; the original state vectors are passed directly to the diffusion model.

**Diffusion Architecture.** The policy generation component is implemented using a modified MLPUnet architecture, inspired by the UNet1D structure (Michael Janner, 2022). Our version retains LayerNorm within the MLP layers for training stability, removes the original attention modules to reduce computational overhead, and replaces the conditional input mechanism (ConID) with an MLP-based residual block. This design simplifies the conditioning pathway while preserving expressivity. The architecture is illustrated in Figure 7.

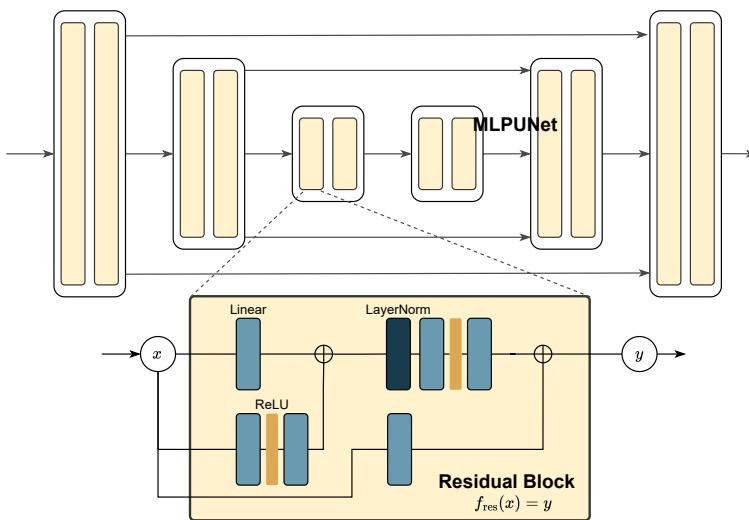

Figure 7: UNet Model architecture in DDGIL. We adopt an MLP module in our DDGIL model, without incorporating any attention modules.

### E.4 COMPUTING RESOURCES

All models are trained and evaluated on an RTX 3090 GPU with 24GB of memory. Both agent and opponent policies are pre-trained using offline datasets. During inference, the diffusion model performs a denoising sampling loop, with the number of steps adjusted based on task complexity.

**Training Cost.** To illustrate computational cost, we report training times for two representative environments: MPE-Push and Atari-Boxing. Training an MPE model takes approximately 10 minutes, while Atari and Badminton require longer due to larger state-action spaces. A summary of these results is provided in Table 10. All experiments are conducted under consistent hardware settings to ensure reproducibility.

| Environment | Dataset Size | Time Cost |
|---|---|---|
| MPE-Push | 50 | 3m 16s |
| | 100 | 10m 34s |
| | 250 | 18m 04s |
| | 500 | 25m 26s |
| Atari-Boxing | 10 | 2m 36s |
| | 50 | 8m 42s |
| | 100 | 18m 31s |
| | 250 | 35m 14s |

Table 10: Training time by dataset size.

**Inference Cost.** We evaluate per-step inference time and memory usage under a fixed setting of 100 episodes against an expert opponent, averaged over three seeds with standard deviation, as shown in Table 11. Compared to BC and DBC, which require only a single forward pass, DDGIL performs a multi-step denoising process, resulting in longer inference time. Compared to DP and DD, DDGIL additionally applies dynamic guidance at each denoising step, further increasing computational cost. For example, in MPE-Push, it requires 67.1 ms per step, slightly higher than DP (32.3 ms) and DD (33.7 ms), primarily due to the added guidance operations within the diffusion sampling process. Memory usage is approximately 0.59 GB for DDGIL, compared to 0.45 GB for DP and DD, and is notably higher than non-diffusion baselines. Despite the increased cost, results in Section 5.3 and Section 5.4 show that DDGIL's enhanced stability and robustness justify the trade-off, particularly under opponent diversity or strategy distribution shifts.

| Environment | Time Cost | Memory Cost |
|---|---|---|
| MPE-Push | 47.1 ms | 0.59 GB |
| Atari-Boxing | 44.5 ms | 0.60 GB |

Table 11: Inference cost: time per step and memory.

## F    INFERENCE DETAILS

To evaluate policy robustness under varying opponent strategies, we categorize adversaries into three skill levels: **Expert**, **Medium**, and **Weak**. We train a reference RL agent for most environments and record reward curves and loss values at each checkpoint. If reward logs are available, we select weights corresponding to the highest, median, and lowest rewards to represent $O_{\text{expert}}$, $O_{\text{medium}}$, and $O_{\text{weak}}$ opponents, respectively. When reward information is unavailable, model checkpoints are chosen based on loss convergence, with the lowest-loss weights serving as the Expert baseline.

In Connect4, we use the pretrained DQN model provided by AgileRL without additional training. In addition to the DQN, AgileRL offers a rule-based agent with three modes: Strong, Weak, and Random. Empirical evaluations reveal that the Strong policy outperforms the pretrained DQN, while the DQN marginally surpasses the Weak policy. Accordingly, we designate:

- **Strong** as Expert
- **DQN** as Medium
- **Weak** as Weak

Although the Weak agent performs similarly to the Medium baseline in some settings, we retain the above classification for consistency.

In the Badminton environment, skill levels are not defined by training performance but rather by real-world players' historical outcomes. The players and data used in this setting derive from professional matches. We select three professional players from the dataset, referred to as K, C, and V, to represent distinct playing styles in our experiments.

Additionally, we employ the RallyNet model as an opponent. RallyNet is a hierarchical offline imitation learning model that reproduces realistic stroke patterns and strategic play styles. We evaluate our method through interactions between RallyNet and other baselines across all six pairwise matchups among the three players.

## G    DATASET INFORMATION

### G.1    DATASET CONSTRUCTION

After training baseline models for each environment in Appendix E, we generate offline datasets by interacting with the expert policies described in Appendix F. Specifically, we collect 500 trajectories for MPE environments (Push, Tag, Spread, Reference) and Classic environments (Connect4, Texas Hold'em), and 250 trajectories for Atari environments (Boxing, Tennis). For Badminton, no additional rollout is required, as the environment includes pre-collected match data for real-world players.

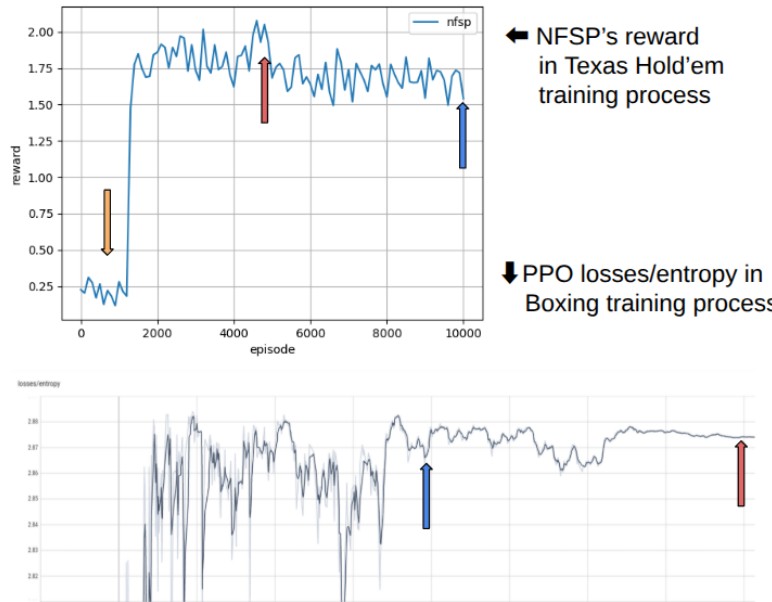

Figure 8: Example training curves used to define skill levels. The top-left plot shows the evaluation reward of NFSP in the Texas Hold'em environment. We selected the highest reward checkpoint as the Expert policy and used checkpoints around 10,000 and 1,000 rewards as Medium and Weak, respectively. The bottom plot shows the loss and entropy of PPO in the Boxing environment; we select the final stage (around 40M steps) as Expert and the mid-stage (around 20M steps) as Medium.

In offline imitation learning, we assume access to a dataset $\mathcal{D} = \{\tau_1, \tau_2, \ldots, \tau_{|D|}\}$. Each trajectory $\tau = \{(S_0, A_0), (S_1, A_1), \ldots, (S_H, A_H)\}$ has a fixed horizon $H$, with $S_t = \{s_t^1, \ldots, s_t^K\}$ and $A_t = \{a_t^1, \ldots, a_t^K\}$ denoting the joint observations and actions of all $K$ agents at timestep $t$.

For implementation, each transition can be further represented as a tuple $\{s_{t-1}^i, a_{t-1}^i, s_t^i, a_t^i\}$, for agent $i$ and timestep $t \leq H$. At $t = 0$, we initialize $s_{-1}^{(i)}$ and $a_{-1}^{(i)}$ as zero vectors to maintain format consistency and enable models to process historical information. The full trajectory can also be viewed as overlapping segments of such tuples:

$$\{s_{t-1}^{(i)}, a_{t-1}^{(i)}, s_t^{(i)}, a_t^{(i)}\}, \quad \{s_t^{(i)}, a_t^{(i)}, s_{t+1}^{(i)}, a_{t+1}^{(i)}\}, \quad \ldots$$

This flexible data structure supports different forms of training batches depending on model requirements:

- **State-action pairs:** $\{s_t^{(i)}, a_t^{(i)}\}$, for standard behavior cloning.

- **Transition tuples:** $\{s_{t-1}^{(i)}, a_{t-1}^{(i)}, s_t^{(i)}, a_t^{(i)}\}$, for models with temporal dependency (e.g., DBC).

- **Trajectory segments:** $\tau^{(i)} = \{s_0^{(i)}, a_0^{(i)}, \ldots, s_n^{(i)}, a_n^{(i)}\}$, $n \in [0, H]$, for sequence-based models or diffusion samplers.

### G.2 DATASET COLLECTION IN DIFFERENT TYPES OF ENVIRONMENTS

Following PettingZoo's environment taxonomy, we organize the datasets under two interaction types: **AEC (Agent Environment Cycle)** and **Parallel**.

- **AEC environments**: At each time step $t$, only one agent receives an observation and acts. The state $s_t$ stores the active agent's observation, while the action $a_t$ is constructed by concatenating the agent's action at $t$ with the opponent's action at $t-1$.

- **Parallel environments**: All agents observe and act simultaneously at each time step. The full state $s_t$ is a concatenation of all agent observations $\{s_t^i\}$, and $a_t$ is a concatenation of all actions $\{a_t^i\}$ in a fixed environment-defined order. Maintaining this order is crucial to ensure proper alignment during training.

During baseline training, we split the dataset by agent name and ordering so that each model only learns from its data. This format is designed primarily to support DBC, which requires $\{s_{t-1}, a_{t-1}, s_t\}$ as input to predict $a_t$. Other baselines use only $\{s_t, a_t\}$ pairs, making this unified structure compatible across methods.

Although no environment rollout is performed for Badminton, the raw match data must be filtered and reformatted. We select a fixed subset of columns as state and action features, structuring the data as $\{s_{t-1}, a_{t-1}, s_t, a_t\}$. Both states and actions contain discrete and continuous variables; discrete features are one-hot encoded and concatenated with continuous values.

Specifically, the state vector includes 1 discrete feature (classes 0–10) and 6 continuous features, resulting in 17 dimensions after encoding. The action vector comprises 1 discrete feature (classes 0–10) and 4 continuous elements, totaling 15 dimensions. These hybrid vectors are used for training after preprocessing.

## H  THE USE OF LARGE LANGUAGE MODELS (LLMS)

During manuscript preparation, we used OpenAI's ChatGPT (GPT-5) to refine wording and improve clarity. The model was occasionally consulted for alternative perspectives in method design and for assistance in presenting mathematical derivations more coherently. We also employed Grammarly for grammar and style checking.

All methodological contributions, theoretical developments, algorithmic designs, and empirical analyses were conceived and validated by the authors, and these tools served only as auxiliary aids for writing and presentation.

## I  LIMITATIONS AND DISCUSSION

While DDGIL achieves strong performance across diverse benchmarks, several limitations are noted. The inference procedure requires recomputing dynamic weights at each denoising step, improving adaptability but incurring higher latency and memory usage than fixed-weight policies. Future work may consider reducing denoising steps or distilling the sampler into a lighter model.

Another limitation is the need to train a separate diffusion model for each agent. This ensures stability by modeling heterogeneous behaviors but scales poorly as agent numbers grow (see ablation experiment D.5). Future work may consider parameter-efficient designs, such as shared encoders with agent-specific decoders, to improve scalability.

The discrepancy signal for dynamic weighting is computed by averaging opponent predictions. While suitable in our benchmarks with relatively homogeneous opponents, this may be less effective under heterogeneous or adversarial settings. Future work may consider weighted aggregation or attention mechanisms for more expressive signals.

Although we did not explicitly consider $w_D < 0.5$, fixed-weight ablations (see experiment D.2) show that performance degrades when weighting is closer to the unconditional update, suggesting that bounding $w_D$ away from the unconditional case aids stability.

Our evaluation does not cover larger multi-agent benchmarks such as MaMuJoCo and SMAC, although the formulation of DDGIL does not rely on benchmark-specific assumptions and may extend to such domains. Further empirical validation remains an important direction for future work.

Finally, adaptation in DDGIL is confined to inference and was not tested under severe distributional shifts. Future work may extend the framework with lightweight online adaptation or meta-learning to improve generalization in such regimes.

