# OpenReview forum: "Diffusion-based Behavior Cloning in Multi-Agent Games via Dynamic Guidance"
_ICLR.cc/2026/Conference — Submitted to ICLR 2026_

### Official Review · Reviewer_QH66 · 2025-10-31

**Soundness:** 2
**Presentation:** 3
**Contribution:** 2
**Rating:** 4
**Confidence:** 4

**Summary:**

This paper tackles the problem of limited adaptability in offline multi-agent imitation learning caused by using fixed diffusion guidance weights. To address this, it introduces a Dynamic Diffusion-Guided Imitation Learning method (DDGIL). The idea is to train separate diffusion models for the primary and opponent agents, and then compute a dynamic confidence weight based on the difference between their noise predictions during the reverse diffusion process. This allows the model to adaptively adjust the guidance strength, improving the policy’s robustness and generalization. Experiments on benchmarks like MPE and Atari show that DDGIL outperforms traditional behavior cloning methods (BC, DBC) and fixed-guidance diffusion approaches (DP, DD) on complex interactive tasks.

**Strengths:**

1. The paper analyzes the problem of poor adaptability caused by fixed diffusion guidance weights in multi-agent environments, which has clear research significance.
2. The experiments cover multiple environments and compare the proposed method with BC, DBC, DP, DD, and offline RL approaches such as OMAR and IDQL. The results are stable and demonstrate the effectiveness of the method.
3. The method is simple to implement and can be easily integrated into existing diffusion policy frameworks.

**Weaknesses:**

1. The methodological novelty is limited, as the design of dynamic guidance is mainly a heuristic extension of the existing classifier-free guidance approach.
2. The multi-agent setting requires training a separate diffusion model for each agent, causing computational and memory costs to grow linearly with the number of agents, which limits scalability.
3. DDGIL remains a pure imitation learning framework without a defined optimality criterion; it relies solely on reconstruction loss and thus cannot learn or improve from failed trajectories.
4. Opponent modeling is only superficial, since each opponent is assigned its own diffusion model, but these models neither make decisions nor perform any game-theoretic reasoning. They only produce noise predictions during inference, which are then used as guidance signals.
5. The paper introduces a noise consistency regularization term that assumes the primary and opponent agents should predict similar noise under the same input. However, it does not explain from a theoretical perspective why enforcing such similarity would help improve policy learning. In multi-agent environments, the primary and opponent agents usually exhibit distinct or even conflicting strategies, so enforcing noise similarity might actually blur these differences and weaken opponent modeling. Moreover, the paper does not include an ablation study to verify the effect or necessity of this regularization term.

**Questions:**

1. The paper presents the method as “offline multi-agent imitation learning,” yet the inference procedure appears to control only the primary agent while opponent models provide guidance signals only. Do you execute policies for all agents during inference? If only a single agent is controlled, is this description accurate?
2. The method also fits the offline data setting and is compared against offline RL baselines. Why not implement and evaluate it within an offline RL framework to assess feasibility, rather than restricting to pure imitation learning?
3. Line 471 states: “DD underperforms, likely due to long-horizon prediction errors.” Conceptually, diffusion-based planning (DD) should be better suited to long-horizon trajectory generation, whereas the proposed dynamic guidance is stepwise and relies on current state information. Why does DD underperform on long horizons while your method improves?
Methods like DD are offline RL algorithms whose objectives depend on returns/values. After adapting them to a reward-free imitation setting, is the comparison fair? Does this modification alter their core optimization objective and thus affect the validity of the comparison?

---

> ### Author Response · Authors · 2025-12-02
> **Response to Reviewer QH66 [1/2]**
>
> Thank you for your careful evaluation of our paper and for providing constructive feedback. Our point-to-point responses to your comments are summarized below.
>
> ---
>
> ### **[W1] The dynamic guidance mechanism appears to offer limited novelty beyond classifier-free guidance.**
>
> We appreciate the reviewer’s observation. Although our method builds on diffusion guidance, the proposed mechanism introduces meaningful innovations for the multi-agent offline imitation learning setting. **Section 4** presents opponent-specific diffusion models and a shared-noise score disagreement that yields a data-driven, interaction-dependent adjustment of guidance strength, fundamentally differing from the static weighting in classifier-free guidance. **Appendix A** further contrasts our formulation with traditional diffusion policies to clarify this distinction. We also highlight these contributions in the revised Conclusion to make the novelty more explicit.
>
>
> ---
>
> ### **[W2] Training a separate diffusion model for each agent may limit scalability.**
>
> We appreciate the reviewer’s observation. We agree that maintaining one diffusion model per opponent introduces a linear cost, and this is a current limitation. In practice, however, offline imitation learning settings typically involve a modest number of opponents, and the environments used in **Section 5** fall within this regime. **Appendix E.4** also reports scalability analysis under increasing numbers of opponents.
>
> ---
>
> ### **[W3] The framework is pure imitation learning and cannot improve from failed trajectories.**
>
> We appreciate the reviewer’s observation. DDGIL is an imitation-based method trained with reconstruction-driven diffusion loss, as rewards and environment interactions are not available in our setting. This limitation is shared by all pure imitation learning frameworks rather than being specific to DDGIL, and the rationale for focusing on the offline imitation regime is stated in **Section 1**. Achieving policy improvement or optimality would require incorporating reward information or value estimation, which would move the method into the offline RL setting and beyond the scope of this work.
>
> ---
>
> ### **[W4] Opponent modeling is superficial because opponent models only provide noise predictions rather than making decisions or performing game-theoretic reasoning.**
>
> We appreciate the reviewer’s observation. The goal of DDGIL is not to perform game-theoretic opponent reasoning but to use opponent-specific diffusion models as contextual guidance for multi-agent imitation learning. These models capture how opponents behave under the same state context and provide score predictions that reflect interaction-dependent structure. During denoising, this information helps the primary policy preserve behavior patterns in the data rather than generate independent decisions. We clarify this design choice in **Section 4.3** and provide additional explanation in **Appendix** A.2 to make the intended role of opponent models explicit.
>
> ---
>
> ### **[W5] The noise consistency regularization lacks theoretical justification and may blur differences between agents.**
>
> We appreciate the reviewer’s observation. The noise consistency term is not intended to align the primary and opponent strategies but to stabilize denoising behavior when models receive shared noise under similar state contexts. This helps maintain coherent latent trajectories and prevents unstable score updates during training. We clarify this motivation in **Section 4.2**, where we explicitly note that the term is used only to stabilize the denoising dynamics and is not designed to reduce or align behavioral differences between agents.

---

> ### Author Response · Authors · 2025-12-02
> **Response to Reviewer QH66 [2/2]**
>
> ### **[Q1] Does the method truly perform multi-agent inference if only the primary agent is controlled?**
>
> We appreciate the reviewer’s question. As described in **Section 4.1**, our inference procedure controls only the primary agent because the setting of this work focuses on learning behavior for a single agent or one side of agents. The opponent diffusion models provide conditional score predictions as guidance during denoising rather than producing actions. This aligns with standard imitation learning scenarios where not all agents are controlled and matches the objectives of our problem formulation.
>
> ---
>
> ### **[Q2] Why not implement the method within an offline RL framework rather than restricting it to imitation learning?**
>
> We appreciate the reviewer’s question. DDGIL is designed for reward-free imitation, and adapting it to offline RL would require introducing rewards, value estimation, or trajectory optimization, which fall outside the problem setting considered in this work. Nevertheless, **Section 5.2** includes comparisons with offline RL baselines such as OMAR and IDQL to provide cross-paradigm reference under the same offline data constraints. We will clarify the method’s intended scope and the rationale for focusing on the imitation setting in the revision.
>
> ---
>
> ### **[Q3] Why does DD underperform in long-horizon settings, and is the reward-free adaptation a fair comparison?**
>
> We appreciate the reviewer’s question. Diffusion-based planning methods such as DD rely on value gradients for trajectory-level optimization, and this advantage is lost in the reward-free imitation adaptation used for comparison. Without value information, DD must fit long behavioral sequences directly, which makes it more prone to compounding prediction errors in dynamic multi-agent environments.
> As noted in the paper, changes in other agents' behavior can shift the interaction context, and long-horizon predictions amplify this mismatch. We follow the reward-free formulation from prior work to preserve consistency across baselines, and we will clarify these limitations and their impact on the comparison in the revision. We also provide an explicit explanation in **Section 5.5** to contextualize DD’s performance drop under the reward-free adaptation.

---

### Official Review · Reviewer_PJmh · 2025-10-31

**Soundness:** 3
**Presentation:** 3
**Contribution:** 2
**Rating:** 4
**Confidence:** 4

**Summary:**

This paper proposes DDGIL, an offline multi-agent imitation learning method that extends diffusion-based behavior cloning with dynamic classifier-free guidance (CFG).  Instead of using a fixed guidance weight (as in Diffusion Policy or Decision Diffuser), DDGIL computes a step-wise adaptive weight based on the disagreement between the primary agent’s conditional prediction and the predictions from auxiliary opponent diffusion models.  The approach modifies only inference, keeping training identical to standard diffusion behavior cloning.  Experiments span multiple domains—MPE, Atari, Classic games, and a real-world badminton simulation—showing that DDGIL typically improves team reward or win rate over strong baselines such as BC, DBC, Diffusion Policy (DP), and Decision Diffuser (DD). Ablations further analyze dataset size, fixed vs. adaptive weights, and scalability in the number of opponents.

**Strengths:**

- The dynamic guidance rule seems lightweight and practical
- The method is tested on diverse benchmarks and real-world badminton data, showing consistent performance gains in many settings
- The paper systematically studies fixed vs. adaptive weights, scalability with opponent count, and shared vs. separate diffusion models, providing good ablation about the trade-offs.

**Weaknesses:**

- The novelty is incremental. The core contribution—a dynamic weighting of classifier-free guidance (CFG)—is conceptually straightforward and builds directly on existing diffusion policy frameworks and opponent modeling ideas already explored in the literature.

- The baseline comparison is limited. The authors only compare against a few adapted single-agent methods, overlooking several established multi-agent imitation learning algorithms such as MAGAIL [1] and MFIQ [2], which would provide a more comprehensive evaluation.

- The theoretical analysis justifies the adaptive weight only locally (through score disagreement) but does not establish any global convergence or stability guarantees. In multi-agent reinforcement and imitation learning, the notion of **local-global consistency**—ensuring alignment between local and global policies in cooperative games—is crucial. The paper lacks a discussion of how DDGIL relates to or preserves such consistency.

- The environments considered in the paper are relatively small in scale. More complex and realistic multi-agent reinforcement learning (MARL) benchmarks, such as **SMACv2**, have recently become standard for evaluating MARL and MAIL algorithms. Including experiments on such environments would strengthen the empirical validation and demonstrate the scalability of the proposed method.


[1] Jiaming Song, Hongyu Ren, Dorsa Sadigh, Stefano Ermon.
“Multi-Agent Generative Adversarial Imitation Learning.” arXiv preprint, Jul 2018. DOI: 10.48550/arXiv.1807.09936.

[2] The Viet Bui, Tien Mai, Thanh H. Nguyen.
“Inverse Factorized Soft Q-Learning for Cooperative Multi-agent Imitation Learning.” In Proceedings of NeurIPS, 2024.

**Questions:**

please address the concerns in the Weaknesses.

**Details Of Ethics Concerns:**

- There are no major ethical concerns associated with this paper. The work focuses purely on algorithmic development and evaluation using simulated or anonymized datasets.
- No human subjects, personal data, or sensitive information are involved in the experiments.
- The proposed method does not raise foreseeable safety or misuse risks beyond standard consideratio

---

> ### Author Response · Authors · 2025-12-02
> **Response to Reviewer PJmh**
>
> Thank you very much for your thoughtful comments and for suggesting areas for improvement. Your comments are first stated and then followed by our point-to-point responses.
>
> ---
>
> ### **[W1] The novelty of the dynamic weighting mechanism may appear limited beyond existing classifier-free guidance methods.**
>
> We thank the reviewer for the feedback. Although our method builds on diffusion-based policies, the contribution is not a direct extension of classifier-free guidance. The proposed formulation restructures guidance for multi-agent interaction rather than adjusting a fixed weighting scheme.
>
> As described in **Section 4.2**, our framework introduces opponent conditional scores that reflect interaction-dependent behavior, which differs from existing single-agent guidance mechanisms. In **Section 4.4**, the adaptive weighting is derived from the disagreement between the primary and opponent predictions and provides a data-driven signal that adjusts guidance strength across denoising steps. This mechanism is important for handling changes in opponent strategies and is not present in prior diffusion policy formulations.
>
> We will add a brief statement in the **Section 6 (Conclusion)** to summarize these innovations.
>
> ---
>
> ### **[W2] The baseline comparison may appear limited because the experiments do not include established multi-agent imitation learning algorithms such as MAGAIL and MFIQ.**
>
> We appreciate the reviewer’s observation. Our baseline choice follows the scope of this work, which is explicitly defined as an offline imitation learning setting. This constraint is stated in the problem formulation in **Section 1** and **Section 3** and reiterated in the experimental setup in **Section 5.2**. The baselines BC, DBC, DP, DD, OMAR, and IDQL all operate under the same offline assumption and do not require interaction with the environment.
>
> Methods such as MAGAIL and MFIQ depend on online adversarial training and continuous environment interaction, which makes them incompatible with the offline regime considered in our study. To maintain fairness and consistency, we therefore compare our approach with algorithms that can be applied within the same offline setting. We also include OMAR and IDQL as offline RL baselines to provide a broader cross-paradigm evaluation, as detailed in **Section 5.2** and **Appendix E**.
>
> ---
>
> ### **[W3] The theoretical analysis focuses on local justification through score disagreement and does not address global convergence or local global consistency in multi-agent settings.**
>
> Our analysis is tailored to diffusion-based offline imitation learning, where the key objective is to describe how the adaptive weight shapes the denoising process. As explained in **Section 4.4**, the theoretical results focus on the relation between score disagreement and the strength of conditional guidance, which directly governs step-wise action generation in our framework.
>
> Global convergence or local global consistency is typically discussed in online multi-agent reinforcement learning, where agents jointly optimize policies through environment interaction. Our setting does not involve online optimization or multi-agent training dynamics, and prior diffusion-based offline imitation learning methods do not rely on such global guarantees. The local analysis we provide is therefore aligned with the assumptions of the offline regime.
>
> ---
>
> ### **[W4] The environments used in the experiments are relatively small compared to large-scale MARL benchmarks such as SMACv2.**
>
> We thank the reviewer for the suggestion. Our experiments focus on MPE, Atari, and Badminton because they expose several core challenges that also arise in larger MARL systems, including high-dimensional observations, interaction-dependent decision making, and diverse multi-agent behavioral patterns. The computational considerations of this offline regime are discussed in **Appendix D** and **Appendix E**.
>
> Although our experiments use smaller environments, DDGIL does not rely on benchmark-specific assumptions, and its training and inference procedures can in principle extend to larger multi-agent domains such as MaMuJoCo and SMAC. We will clarify this general applicability in the revision, and we note that this point is also discussed in the Limitations section in **Appendix I**. Evaluating DDGIL on large-scale MARL benchmarks remains a meaningful direction for future work.

---

### Official Review · Reviewer_Ydk8 · 2025-11-01

**Soundness:** 2
**Presentation:** 2
**Contribution:** 2
**Rating:** 2
**Confidence:** 4

**Summary:**

The paper studies behavioral cloning in multi-agent settings. The paper's main idea is to apply diffusion models to generate actions based on observations of agents. Given the presence of multiple agents, the paper selects an agent as a primary agent and the others as opponents, diffusion-based policy learning is then developed accordingly to learn local policies for individual agents. The paper then introduces a dynamic weight adjustment based on the alignment between the primary agent and its opponents to interpolate between conditional and unconditional noises in diffusion models. Experiments on various tasks including MPE and Atari games show the proposed BC-based multi-agent imitation learning performs better than baselines such as BC, diffusion BC, diffusion policy, and decision diffusion.

**Strengths:**

1. Experiments show promising results of the proposed method.

**Weaknesses:**

***The justifications for the proposed model’s design choices are insufficient, particularly regarding how diffusion models are utilized for multi-agent action generation:

1. Choice of Primary Agent: The paper does not explain how this agent is selected or how such a choice impacts imitation performance, especially in environments with heterogeneous agent roles.

2. Opponent Diffusion Modeling: Opponent actions are conditioned on the primary agent’s state, which lacks rationale. In standard multi-agent settings, each agent’s actions should depend on its own observations and policies, not solely on the primary agent.

3. Shared Noise Across Agents: The method enforces identical noise inputs for all agents’ diffusion models, yet the paper does not justify why this condition is necessary or meaningful.

4. Lack of Multi-Agent Interaction Modeling: The approach does not clarify how cooperative or competitive interactions are captured. Understanding joint behavior dynamics is crucial in multi-agent learning

*** Experimental Evaluation
1. Limited Baselines: Several established single-agent offline imitation learning methods could be used as baselines by training policies per agent individually. Including such baselines would provide a more comprehensive and convincing comparison.

**Questions:**

1. Please address my points on weaknesses.

2. Can your method apply for complex domains such as MaMujoco and SMAC? If so, can you please show some empirical results?

---

> ### Author Response · Authors · 2025-12-02
> **Response to Reviewer Ydk8**
>
> We thank the reviewer for the constructive comments and positive feedback on our paper. Our point-to-point responses are summarized below.
>
> ### **[W1] The paper does not explain how the primary agent is chosen or how that choice affects performance in heterogeneous settings.**
>
> We thank the reviewer for the helpful comment. As described in **Section 4.1**, a diffusion model is trained for each agent, and the primary agent is the one selected for control during inference. This choice is user-specified and not related to role-specific information. Since all agents share the same architecture and training procedure, our method applies to environments with heterogeneous roles without introducing bias or affecting imitation performance. We will clarify this at the end of the first paragraph in **Section 4.1**.
>
> ---
>
> ### **[W2] Conditioning opponent models only on the primary agent’s observation is insufficiently justified relative to standard multi-agent decision-making.**
>
> We appreciate the reviewer’s comment. As clarified in **Section 4.2**, we do not assume that opponents base their actual policies solely on the primary agent's observation. The opponent diffusion models use *sᴳ* only to provide a consistent contextual reference during training, enabling their denoising predictions to be compared meaningfully with those of the primary agent. These models do not generate actions during inference and serve only as auxiliary predictors whose discrepancies with the primary model form the confidence signal used in our dynamic guidance mechanism. This design supports adaptive conditional weighting without altering opponent behavior or requiring assumptions about their decision-making processes. We will add a short clarification in the middle of **Section 4.2** to make this intent explicit.
>
>
> ---
>
> ### **[W3] The requirement that all diffusion models share identical noise lacks a clear motivation.**
>
> Shared noise ensures that the primary and opponent diffusion models follow the same denoising trajectory so that their discrepancies reflect differences in predicted behavior rather than noise-induced randomness. If independent noise were used, the term *d = ‖εᴮ − εᴳ‖* would be dominated by stochastic variation and would no longer provide a reliable confidence signal for dynamic guidance. This shared-noise design allows disagreement to capture meaningful behavioral differences across agents. We will add a brief clarification in **Section 4.2** to state this motivation clearly.
>
> ---
>
> ### **[W4] The method does not clearly describe how cooperative or competitive interactions are captured.**
>
> We appreciate the reviewer’s comment. Although DDGIL does not explicitly model a joint policy, it captures cooperative and competitive interactions through shared trajectory data and opponent-aware diffusion modeling. As described in **Sections 4.2 and 4.3**, the primary and opponent diffusion models are trained on the same multi-agent trajectories, incorporating a latent consistency term, which enables them to learn co-occurring behavioral patterns. During inference, opponent models provide conditional predictions under the same context, and their discrepancy with the primary model yields the confidence signal for dynamic guidance. This enables the primary policy to adapt to aligned or conflicting behaviors and implicitly reflects cooperative or competitive dynamics.
>
> ---
>
> ### **[W5] Several standard per-agent offline imitation learning baselines are missing from the evaluation.**
>
> We appreciate the reviewer’s suggestion. As noted in **Section 5.2**, we already include per-agent offline imitation learning baselines by training Behavior Cloning (BC) and diffusion-augmented BC (DBC) separately for each agent. These represent standard single-agent offline IL approaches and serve as appropriate baselines for comparison.
>
> ---
>
> ### **[Q2] It is unclear whether the method applies to larger benchmarks such as MaMuJoCo or SMAC and whether empirical results can be provided.**
>
> We thank the reviewer for the question. Our experiments focus on Badminton, Atari, and MPE because they span key challenges also present in complex domains such as MaMuJoCo and SMAC, including high-dimensional control, visual decision-making, and multi-agent coordination with interaction variability. These environments allow us to assess DDGIL across diverse modalities while keeping computational cost manageable, as analyzed in **Appendix D** and **Appendix E**.
>
> In principle, DDGIL can extend to larger multi-agent systems such as MaMuJoCo and SMAC, as its training and inference do not rely on benchmark-specific assumptions. We will clarify this general applicability in the revised manuscript, and we note that this point is also discussed in the Limitations section in Appendix I. Evaluating DDGIL on large-scale MARL benchmarks is an important direction for future work.

---

### Author Response · Authors · 2025-12-03
**Summary of Responses for Paper 20695**

Dear (new) ACs and SACs,

We greatly appreciate your efforts in coordinating our submission under these unusual circumstances. Since the discussion phase concluded before we could submit a full set of responses, we provide a one-sentence summary for each point of feedback below. All clarifications have been added to address concerns regarding novelty, prompts, and datasets. Please refer to the Summary of Revision and the full responses for details.

---

## Reviewer Ydk8

### **[W1] Choice of Primary Agent**

We define the primary agent in **Section 4.1** and will add a short clarification specifying that its selection is user-defined and role-independent.

### **[W2] Opponent Diffusion Modeling**

We explain the conditioning choice and the auxiliary role of opponent models in **Section 4.2** and will add a clarifying sentence in the middle of that section.

### **[W3] Shared Noise Across Agents**

We describe the shared-noise design in **Section 4.2** and will add a brief statement on its motivation.

### **[W4] Interaction Modeling**

We describe how interaction patterns are captured through shared trajectories and opponent-aware diffusion modeling in **Sections 4.2 and 4.3**.

### **[W5] Missing per-agent offline IL baselines**

We already include per-agent BC and DBC baselines in **Section 5.2**.

### **[Q2] Applicability to MaMuJoCo and SMAC**

We discuss computational considerations in **Appendix D** and E** and provide a clarification on the method’s applicability to larger benchmarks in the Limitations section of **Appendix I**.

---

## Reviewer PJmh

### **[W1] Dynamic weighting novelty**

We clarify in **Sections 4.2 and 4.4**, and in the revised **Sections 6 (Conclusion)**, how DDGIL reformulates diffusion guidance for multi-agent interaction.

### **[W2] Missing multi-agent IL baselines**

We state in **Sections 1, 3, and 5.2** that the setting is strictly offline and compared only with methods that operate under this same assumption.

### **[W3] Lack of global convergence analysis**

We explain in **Section 4.4** that our analysis focuses on local score behavior appropriate for offline diffusion IL, where global convergence guarantees do not apply.

### **[W4] Experimental environments are small**

We discuss computational constraints in **Appendices D and E**, note general applicability in the revision, and address this point in the Limitations section in **Appendix I**.

---

## Reviewer QH66

### **[W1] Novelty of dynamic guidance**

We clarify in **Sections 4** and **Appendix A**, and further emphasize in the revised **Section 6 (Conclusion)**, how DDGIL restructures diffusion guidance for multi-agent interactions.

### **[W2] Scalability of per-agent diffusion models**

We discuss the applicability conditions in **Section 5** and **Appendix E.4**, and explicitly clarify this limitation in the revision.

### **[W3] Pure imitation learning cannot improve from failures**

We state in **Section 1** that DDGIL is a reward-free imitation method, and policy improvement lies outside the offline IL scope.

### **[W4] Opponent modeling appears superficial**

We explain in **Section 4.2** that opponent models provide contextual score predictions rather than strategic decisions and clarify this design intent in the revision, and clarify in **Section 4.3** (with additional details in **Appendix A.2**) that opponent models provide contextual score predictions rather than strategic decisions.

### **[W5] Noise consistency regularization lacks justification**

We clarify its role in stabilizing denoising dynamics in **Section 4.2**, with further explanation added in **Appendix A.2**.

### **[Q1] Is this truly multi-agent inference if only the primary agent is controlled?**

**Section 4.1** explains that our setting focuses on controlling one agent while using opponent models only as guidance.

### **[Q2] Why not use an offline RL framework?**

We clarify in **Section 1** and **Section 5.2** that DDGIL targets reward-free imitation and compare with offline RL baselines under the same constraints.

### **[Q3] Why does DD underperform in long-horizon settings, and is the reward-free adaptation fair?**

We explain the cause in **Section 5.5** and note in the revision that the reward-free adaptation removes value gradients essential to DD’s long-horizon optimization.

---

### Meta-Review · Area_Chair_vyPM · 2026-01-06

**Summary:**

The paper proposes Diffusion Dynamic Guidance Imitation Learning (DDGIL) to address limitations of fixed-weight classifier-free guidance in offline multi-agent imitation learning. The core approach involves training auxiliary diffusion models for opponents and utilizing the prediction discrepancy under shared noise to dynamically adjust guidance weights during inference. While reviewers acknowledged the clear motivation and solid experimental results on MPE, Atari, and Badminton, the consensus leans towards rejection. The primary reasons are the limited methodological novelty, viewed as an incremental heuristic over CFG, and scalability concerns due to the linear computational cost regarding the number of agents. Despite a thorough rebuttal, the structural limitations regarding scalability and the lack of evaluation on complex MARL benchmarks like SMACv2 remain significant barriers to acceptance.

**Reviewer Concerns:**

The authors successfully addressed concerns regarding baseline selection by clarifying why online or adversarial baselines like MAGAIL are inappropriate for their strictly offline setting. The comparison with offline RL baselines and per-agent BC/DBC was deemed fair after this clarification. Furthermore, the rationale for shared noise to ensure discrepancies reflect behavioral differences rather than randomness and the role of opponent models as auxiliary context were well-explained, resolving specific confusion from Reviewers Ydk8 and QH66.
However, outstanding concerns remain regarding marginal novelty and scalability. Reviewers PJmh and QH66 maintained that the dynamic weighting mechanism is a straightforward heuristic extension of Classifier-Free Guidance with thin theoretical contribution. The requirement to train and infer separate diffusion models for every opponent introduces an O(N) computational and memory cost, which acts as a critical bottleneck limiting practical impact. Additionally, the absence of large-scale benchmarks like SMACv2 weakens empirical claims given these scalability concerns, as current experiments on MPE and Atari are considered relatively small-scale.

**Reviewer Scores:**

Reviewer Ydk8 (Score: 2) might slightly appreciate the clarifications on baselines and design choices, but fundamental concerns about the method's complexity likely prevent a significant score increase.
Reviewer PJmh (Score: 4) found the work incremental, and the rebuttal did not alter this view, keeping the score borderline.
Reviewer QH66 (Score: 4) raised valid concerns about scalability which were acknowledged by authors as limitations rather than resolved.

---

### Decision · Program_Chairs · 2026-01-26

Reject